# Deep-learning models reveal how context and listener attention shape electrophysiological correlates of speech-to-language transformation

Andrew J. Anderson[1,2,3,4]*, Chris Davis[5], Edmund C. Lalor[4,6,7]

**1** Department of Neurology. Medical College of Wisconsin, Milwaukee, Wisconsin United States of America, **2** Department of Biomedical Engineering. Medical College of Wisconsin. Milwaukee, Wisconsin United States of America, **3** Department of Neurosurgery. Medical College of Wisconsin. Milwaukee, Wisconsin United States of America, **4** Department of Neuroscience and Del Monte Institute for Neuroscience, University of Rochester, Rochester, New York, United States of America, **5** Western Sydney University, The MARCS Institute for Brain, Behaviour and Development, Westmead Innovation Quarter, Westmead, New South Wales, Australia, **6** Department of Biomedical Engineering, University of Rochester, Rochester, New York, United States of America, **7** Center for Visual Science, University of Rochester, Rochester, New York, United States of America

\* andanderson@mcw.edu

**Data Availability Statement:** Raw EEG data are publicly available from Broderick et al. 2019, doi:10.5061, https://doi.org/10.5061/dryad.070jc. Preprocessed EEG data, models and code to

## Abstract

To transform continuous speech into words, the human brain must resolve variability across utterances in intonation, speech rate, volume, accents and so on. A promising approach to explaining this process has been to model electroencephalogram (EEG) recordings of brain responses to speech. Contemporary models typically invoke context invariant speech categories (e.g. phonemes) as an intermediary representational stage between sounds and words. However, such models may not capture the complete picture because they do not model the brain mechanism that categorizes sounds and consequently may overlook associated neural representations. By providing end-to-end accounts of speech-to-text transformation, new deep-learning systems could enable more complete brain models. We model EEG recordings of audiobook comprehension with the deep-learning speech recognition system Whisper. We find that (1) Whisper provides a self-contained EEG model of an intermediary representational stage that reflects elements of prelexical and lexical representation and prediction; (2) EEG modeling is more accurate when informed by 5-10s of speech context, which traditional context invariant categorical models do not encode; (3) Deep Whisper layers encoding linguistic structure were more accurate EEG models of selectively attended speech in two-speaker "cocktail party" listening conditions than early layers encoding acoustics. No such layer depth advantage was observed for unattended speech, consistent with a more superficial level of linguistic processing in the brain.

reproduce results are publicly available at: https://osf.io/7vzqb/.

**Funding:** AJA and ECL were supported by a Del Monte Institute Pilot Project Program grant. This research was funded in part by the Advancing a Healthier Wisconsin Endowment (AJA). AJA received salary from the Advancing a Healthier Wisconsin Endowment. ECL was supported by National Science Foundation CAREER award 1652127. AJA and ECL received salary from National Science Foundation award 1652127. CD was supported by an Australian Research Council DISCOVERY award DP200102188. The funders had no role in study design, data collection and analysis, decision to publish, or preparation of the manuscript.

**Competing interests:** The authors have declared that no competing interests exist.

## Author summary

Most people effortlessly can understand different speakers with distinct voices and accents, no matter whether they whisper, shout or are happy or sad. This effortlessness belies the remarkable computational challenge that our brains solve to transform such variable speech sounds into words. It is believed that our brains deal with this variability by categorizing speech sounds into a flow of phonemes and/or syllable units that are consistent, no matter how a word is spoken. Compelling supporting evidence has come from electrophysiological recordings of brain activity–colloquially known as brain waves—taken as people listen to speech. Scientists have trained computational models to predict brain wave fluctuations that correlate with sequences of phoneme categories. However, modeling only phoneme categories may miss key stages in the recognition process, including how sounds are mapped to phonemes, and phonemes to words. New deep learning speech models that are trained to recognize a diverse range of speech and speakers may offer new opportunities to provide more complete accounts of brain activity. This article reveals that these models indeed predict hitherto unexplained fluctuations in speech brain waves that reflect elements of sub-words and words, and shows that fluctuations are context sensitive, which may reflect the brain anticipating upcoming speech.

## Introduction

The apparent ease with which the human brain transforms speech sounds into words belies the complexity of the task. This complexity is due in large part to speech variability—each time a word is spoken, the sound is different. Speech variability is most striking in extreme cases such as when people have unfamiliar accents, shout, whisper or sing, but is always present to some degree, even when the same person repeats the same phrase [1,2]. How brains transform such variable speech sounds into language is a key unresolved question in cognitive neuroscience. By enabling temporally precise estimates of brain activity, electrophysiological measures such as scalp EEG have provided evidence that the brain transforms natural continuous speech into words as a cascading process with abstract categorical speech units such as phonemes or their articulatory features serving as intermediary pre-lexical representations [3–6]. The transformation to phoneme sequences prospectively enables the brain to distinguish word identity, independent of the speaker, intonation, volume, word meaning and so on. To support this, researchers have typically revealed how EEG models of speech comprehension are improved when representing the speech stimulus as a time-series of categorical phoneme feature vectors in addition to the audio signal (e.g. [3]). However, as observed by [7]: (1) The phoneme predictive advantage revealed by [3] can more parsimoniously be explained by phoneme timing rather than phoneme identity or articulatory structure, and phoneme timing can be approximated by estimating the derivative of speech energy. (2) As a broader issue, it is desirable to produce more complete models of the speech recognition process, because categorical models typically do not specify the computational mechanism that categorizes variable speech sounds, nor how sub-word categories are derived from audio data. They therefore may overlook key components of the speech recognition process in the brain.

By modeling speech recognition end-to-end from audio to words, with human-like accuracy, recent deep artificial neural networks such as Whisper [8] present new opportunities to alleviate the second concern above and potentially provide a new window on speech comprehension in the brain. Critically, different to categorical speech models, intermediary representations within Whisper are a learned function of the audio spectrogram that has been

optimized to reproduce word-level speech transcriptions made by human annotators. Thus, Whisper might not only discover intermediary phoneme representations but also learn how to exploit phonetic and lexical context in service of speech recognition, which in turn might model new and/or known electrophysiological correlates of natural speech comprehension (e.g. [3,7,9,10,11,12]). Indeed, recent empirical studies [13,14] of the self-supervised speech model Wav2Vec2 [15] have demonstrated its sensitivity to phonological context and lexical knowledge, and a range of different speech models have been observed to encode phonemes [16], syntax and semantics to some degree [17].

Operationally, Whisper turns continuous audio speech into categorical word units via a succession of intermediary transformations that take place within an "Encoder-Decoder" Transformer architecture [18]. The Encoder module prepares input speech spectrograms for word decoding. This is achieved by re-representing each spectrogram timeframe as a "contextualized" weighted average of itself and all other time frames within a 30s window. The contextualization process is repeated across multiple intermediate layers, each feeding forward into the next. The output of the final layer, which is the output of the entire Encoder module is a time-series of contextualized speech vectors that are fed as one 30s chunk into the Decoder. The Decoder is also a multilayer feed-forward Transformer network, which then transcribes the encoded speech into a series of discrete word units in the same or a different language. This proceeds as an iterative process, with the decoder predicting the identity of the next word based on the encoded speech and any words it has previously decoded. Thus, the decoder closely resembles next-word-prediction language models such as GPT-2 [19], but with the additional access to contextually encoded speech.

Here, we model audiobook speech EEG recordings with Whisper's Encoder module, with an eye to identifying whether Whisper affords a predictive advantage over traditional acoustic and phoneme measures, which we find, and then characterizing what underpins this advantage. We based our EEG analyses solely on Whisper's Encoder on the assumption that it plays the key role in transforming audio speech into a linguistic form to help the Decoder to predict word identity, otherwise the Encoder would be redundant. We further assumed that Encoder representations would become more linguistic with layer depth due to the feedforward architecture and contextualization process (see Methods for more details on these assumptions). However, the degree to which the learned linguistic forms reflect sub-words, words, or even semantics to successfully interface with the word decoder, and which of these features contribute to EEG models requires further investigation to estimate, which we undertook. To ease descriptions in the forthcoming text, we refer to the transformation performed by Whisper's Encoder as a speech-to-language transformation–with the proviso that language is ambiguously defined–and the conclusion we finally reach is that Whisper learns a mixture of sub-word and word structure, in part reflecting lexical predictions, and this helps to model EEG.

To characterize the correlation between EEG and Whisper we first examined how EEG predictions vary across model layers, as has been examined with language models in fMRI, MEG or ECoG [20–29], and more recently speech models [29,30–33]). We analyzed both audiobook comprehension EEG recordings made in: (1) single-speaker listening conditions, and (2) an experimental "cocktail-party" scenario where participants listened to two concurrent audiobooks but paid attention to only one [34]. Extrapolating from [9]'s finding that correlates of lexical processing selectively reflect only the attended audiobook, we hypothesized the same would be true for deeper-more linguistic Whisper layers, whereas correlates of lower-level acoustics would remain for both attended and unattended speech, albeit to different degrees [34–37]. To estimate which features of Whisper drove EEG prediction we performed comparative analyses with a pure language model (GPT-2 [19]) and also two self-supervised speech models (Wav2Vec2 [15] and HuBERT [38]) which appear to induce aspects of lexical

semantics without access to text annotations [31,39]. To probe for EEG correlates of sub-word representation we tested whether shuffling the order of Whisper vectors within words disrupted modeling accuracy. To establish that EEG responses reflect Whisper's contextualized speech transformations, we tested whether limiting Whisper's access to context compromised modeling accuracy, as has been studied in language and fMRI [20].

## Results: overview

We first reanalyzed a dataset of publicly available Electroencephalographic (EEG) recordings [40] taken from 19 subjects as they listened to ~1hour of an audiobook (The Old Man and the Sea [41]). We hypothesized that the internal representations of Whisper–which reflect a graded transformation of spectral speech input into a linguistic form–would more accurately predict EEG responses than spectral features of the speech, or derivatives thereof. This was because Whisper, like the human brain, is adapted to transform speech to language. Because it is well established from N400 studies that EEG is sensitive to language [42–43] and in particular word expectation [12,44], we ran a battery of control analyses to gain confidence that Whisper was not re-explaining established correlates of language processing. These control tests were primarily based on estimates of lexical (word) surprisal (how unexpected a word is based on prior context), though we later reference findings to the internal states of the language model used to generate next word expectations (GPT-2 [19]).

## Natural speech EEG recordings reflect the linguistic transformation of acoustic speech

To establish how accurately different Whisper layers predicted EEG and determine if they complemented acoustic and lexical models of speech, we ran a series of cross-validated multiple regression analyses (see **Fig 1** and **Methods** for details). In each analysis, a model to EEG mapping was fit to each individual's data. To estimate the complementary predictive value of different models, EEG prediction accuracies derived from model combinations were contrasted with those from constituent models. Model combinations are referred to as Unions in forthcoming text and/or completely specified as [Whisper LX Control], where LX corresponds to Layer number X and the square brackets indicate concatenation of Whisper features with coincident control model features. Control models were: (a) The audiobook speech envelope concatenated with its first half wave rectified derivative (abbreviated as Env&Dv). (b) An 80 channel Log-Mel Spectrogram corresponding to Whisper's input and computed by Whisper's preprocessing module. (c) Lexical Surprisal–a measure of how unexpected each word is. Surprisal values were computed using GPT-2 to anticipate the identity of each forthcoming word based on up to 1024 prior words, which is represented as a long vector of probability values linked to each word in GPT-2's dictionary. Lexical surprisal values for individual words were computed as the negative log probability estimates associated with those words. A time-series representation was then constructed by aligning lexical surprisal "spikes" to word onset times (relative to the EEG time-line) and setting all other time-series values to zero.

To first establish whether Whisper complemented each individual control model, we evaluated whether pairwise [Whisper LX Control] scalp-average EEG predictions were more accurate than predictions derived from the Control alone. This evaluation used signed-ranks tests (one-tail, 19 subjects), with p-values corrected across layers for multiple comparisons using False Discovery Rate (FDR). There were two principal findings (both illustrated in **Fig 2** **Top Row**): (1) Whisper vectors from L1-L6 strongly complemented all Controls in prediction. For example, [Whisper L6 Env&Dv] had an accuracy of Mean±SEM r = 0.056±0.004 which was greater than Env&Dv with Mean±SEM r = 0.041±0.003 (please see **Fig 2** for sign rank test

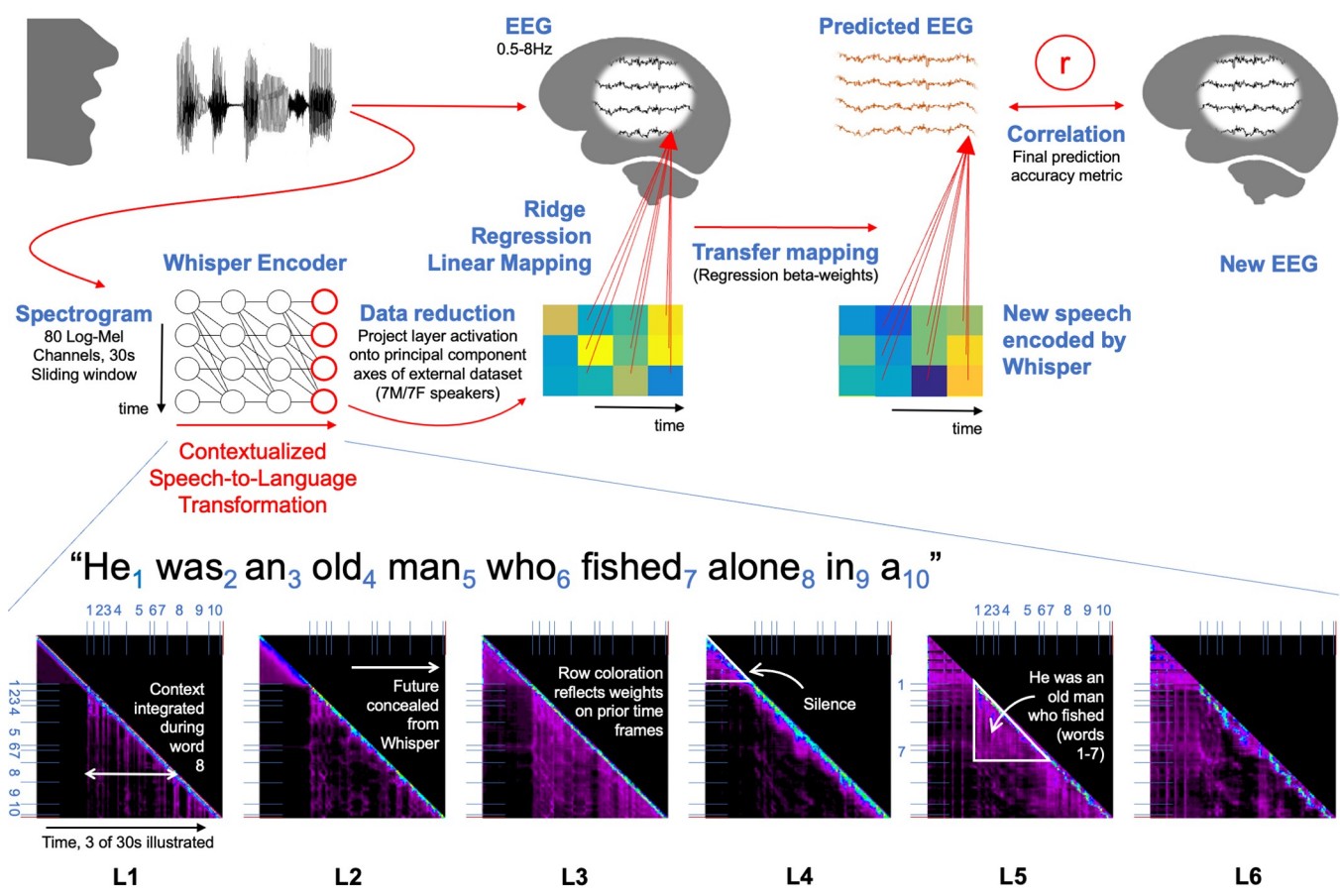

**Fig 1. Predicting Natural Speech EEG Recordings with a Contextualized Speech Model.** EEG recordings of audiobook comprehension were analyzed. The audiobook waveform was processed through a pre-trained deep-learning speech model (Whisper-base). A sliding window approach was applied to feed the model up to 30s of prior speech audio waveform, which was then re-represented as an 80 channel Log-Mel Spectrogram. The Spectrogram is then fed-forward through successive layers of a Transformer Encoder artificial neural network via an initial convolutional layer. This entire process can be considered to implement a graded transformation of input speech to a contextualized more linguistic representation. At each transformer layer, input time-frames are contextualized via a self-attention computation that re-represents input frames according to a weighted average of themselves and preceding frames. The bottom row illustrates a summary of self-attention weightings computed at each layer for the first 3s of the audiobook stimulus. Attention weights (all positive values) relating each time frame to each previous timeframe are illustrated as the colored shading on each matrix row. Specifically, points along the diagonal correspond to attention weights at any timepoint t (from 0 to 3s) unrelated to preceding context. Meanwhile, points to the left of the diagonal correspond to the attention weights applied to preceding frames to re-represent the current time frame. The wealth of color to the left of the diagonal in layers 3, 5, and 6 demonstrates the importance of prior context in Whisper's operation. The range of values in each matrix is L1: [0 0.16], L2: [0 0.15], L3: [0 0.09], L4 [0 0.13], L5 [0 0.11], L6 [0 0.09], with color intensity reflecting weight strength. The self-attention computation is illustrated in detail in **Fig A in S1 Text**, and self-attention weight maps computed in the eight attention heads that were averaged to generate the visualization above are in **Figs B and C in S1 Text.** Whisper layer outputs were used to predict co-registered EEG data in a cross-validated multiple regression framework (this is illustrated above for only the final layer output). To reduce computational burden, Whisper vectors were reduced to 10 dimensions by projection onto pre-derived PCA axes (computed from different audiobook data, see also **Fig D in S1 Text**), and both EEG and model data were resampled at 32Hz.

statistics). (2) EEG prediction accuracies became successively stronger as a function of layer depth. The Mean±SEM Spearman correlation coefficient between prediction accuracy and layer depth (0 to 7) across participants was 0.77±0.07. The set of layer depth vs prediction accuracy correlation coefficients (for the 19 participants) were significantly greater than 0 (Signed-rank Z = 3.7924, p = 1.5e-4, n = 19, 2-tail). This provided evidence that deeper and more linguistic Whisper layers were the strongest EEG predictors.

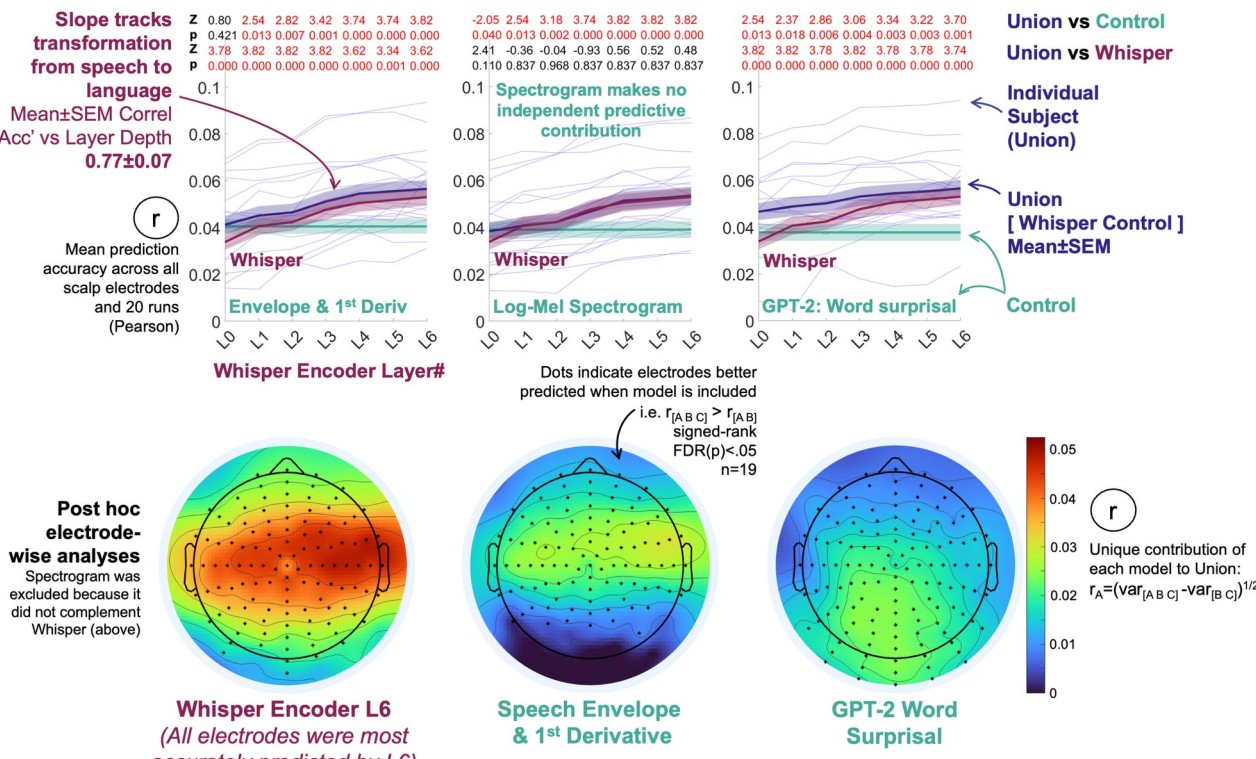

**Fig 2. EEG is more accurately predicted by a contextualized speech model than standard acoustic or lexical surprisal representations, with accuracy increasing with model layer depth. Top row line plots:** The speech model (Whisper) complemented Speech Envelope-based measures, a Log-Mel Spectrogram (Whisper's input) and lexical surprisal in predicting EEG data. Models were considered to be complementary if scalp-average EEG prediction derived from concatenating model pairs (Union = [Whisper, Surprisal]) were more accurate than constituent models (signed-ranks tests, Z and FDR corrected p-values are displayed at the top of each plot). Whisper Layers 1 to 6 complemented every competing predictor. Lexical surprisal and the envelope-based predictors, but not the spectrogram also made independent contributions to prediction. Corresponding signed-rank test statistics (Z, FDR(p)) are the red and black numbers at the top of each plot. Individual-level scalp-average results for Env&Dv, Log-Mel Spectrogram and Word Surprisal are in **Fig E in S1 Text**. **Bottom row scalp maps:** Post hoc electrode-wise analyses mapped scalp-regions that were sensitive to the speech-to-language model. Each electrode was predicted using the Union of Whisper Layer 6, the Envelope-based measures and Lexical Surprisal (because all three had made independent predictive contributions in the primary scalp-average analyses, unlike the spectrogram which was excluded). Whisper's independent contribution was estimated by partitioning the variance predicted by the three-model union (including Whisper) minus the variance predicted by a joint model excluding Whisper (Envelope and Surprisal). The square root of the resultant variance partition was taken to provide a pseudo-correlation estimate on the same scale as other results, as is a common procedure [31,45]. Electrodes that are typically associated with low-level acoustic processing were especially sensitive to Whisper. This is visible as the red band that straddles central scalp from ear to ear in the leftmost scalp map.

To identify which Control models complemented Whisper, we evaluated whether Union prediction accuracies were greater than constituent Whisper layers (Signed ranks tests, one-tail, 19 subjects). This revealed that both Env&Dv and Lexical Surprisal (**Fig 2** Top Left and Top Right), but not the Spectrogram (Whisper's input) uniquely predicted elements of the EEG signal (**Fig 2** Top Middle). In sum, these results provide evidence that EEG strongly reflects the transformation of speech to language encoded by Whisper.

To next establish which electrodes reflected speech-to-language transformations, we ran a series of post hoc electrode-wise prediction analyses that combined Whisper's final most linguistic layer (L6 –which as it turned out also generated the most accurate predictions of each electrode) with both Env&Dv and Lexical Surprisal. **Fig F in S1 Text** presents supporting electrode-wise analysis of all Whisper Layers, finding L6 coded almost all of the information that was valuable for predicting EEG, and earlier layers predicted a similar array of electrodes with lower accuracy. For completeness, **Fig G in S1 Text** presents a supporting analysis showing

that Whisper Layers 1–6 model EEG features predicted by a phoneme articulation model [3], though these same EEG features may reflect acoustic measures [7]. The Log-Mel Spectrogram was excluded from this and all further analyses because it had made no independent predictive contribution in our initial analyses (**Fig 2 Top Row**).

To estimate whether Whisper uniquely contributed to predicting an individual electrode, signed ranks tests (19 subjects, one-tail) were used to evaluate whether electrode predictions derived from three model Union: [Whisper L6 Env&Dv Lexical Surprisal] were more accurate than when Whisper was excluded (i.e., [Env&Dv Lexical Surprisal]). P-values across the 128 channels were FDR corrected for multiple comparisons. Analogous tests were deployed to reveal each electrode's unique sensitivity to Env&Dv (e.g. Union vs [Whisper Lexical Surprisal]) as well as Lexical Surprisal (Union vs [Whisper Env&Dv]). To estimate the relative contribution of each model to predicting each electrode we deployed a predicted-variance partitioning approach [31,45]. For instance, Whisper's independent predictive contribution was estimated as the variance predicted by the Union minus the variance predicted when Whisper was excluded, i.e. $r^2_{[Whisper\ Env\&Dv\ Lexical\ Surprisal]} - r^2_{[Env\&Dv\ Lexical\ Surprisal]}$. The square root of the resultant variance partition was taken to provide a pseudo-correlation estimate, as is a common practice. In the instance of negative variance partitions (as can arise from overfitting in regression, despite regularization), pseudo correlation estimates were zeroed. Scalp maps of pseudo correlation variance partition estimates overlaid with the outcomes of the above signed ranks tests are illustrated in **Fig 2 Bottom Row**.

The electrode-wise analyses revealed two principal findings: (1) Linguistic Whisper L6 dominated prediction in bilateral scalp electrodes that are traditionally associated with low-level speech processing (and also captured here in part by Env&Dv). (2) EEG correlates of Lexical Surprisal were distinct and observed in centroparietal electrodes that are traditionally associated with the N400. In sum, these results link the new EEG correlates of speech-to-language transformation in Whisper to bilateral scalp electrodes which are typically associated with processing acoustic speech or speech units.

## EEG correlates of speech-to-language transformation are attention-dependent in cocktail-party environments

Given that Whisper accurately predicts scalp EEG in traditional speech processing regions, it was possible that Whisper's predictive advantage was gained by providing a high-quality brain-like filter of concurrent surface-level spectral features. We reasoned that one compelling test of this would be to modulate the depth of linguistic processing on the listener's side, and test whether the EEG data's sensitivity to Whisper was likewise modulated.

One effective way of modulating a listener's engagement to speech is via selective attention. In so-called cocktail-party scenarios it is widely appreciated that listeners can hone in on a single speaker whilst ignoring others [34–36]. Indeed, the electrophysiological bases of selective attention in multi-talker environments have been thoroughly investigated and there is general agreement that unattended speech is processed at a more superficial level than attended speech [9,34–37]). For instance, listeners have low ability to accurately report on the unattended speech content, and N400-like lexical expectation responses disappear for unattended speech [9] whilst traces of low-level acoustic speech processing remain, albeit at a diminished level [34]. We therefore hypothesized that correlates of Whisper observed in **Fig 2** would be pronounced for the attended speaker in deep layers, but would dwindle or disappear for unattended speech, whereas acoustic correlates would remain to some degree. In particular, we hypothesized that the predictive advantage associated with Whisper layer-depth (the slope across layers in **Fig 2**) would flatten out for unattended speech.

To test the above hypothesis, we reanalyzed a publicly available Cocktail Party EEG dataset (https://doi.org/10.5061/dryad.070jc [40]) where EEG was recorded as listeners were played 30mins of two audiobooks presented simultaneously [34]. The two audiobooks–"20,000 Leagues under the Sea" [46] and "Journey to the Center of the Earth" [47] were narrated by different male speakers, and were presented via headphones, one to the left ear and the other the right. Participants were tasked with selectively attending to one audiobook (in one ear) across the entire experiment. We analyzed 15 participants who paid attention to "20,000 Leagues. . ." and 12 who paid attention to "Journey. . .". Audiobook presentation was split in thirty runs of 1 min each that were separated by brief breaks to mitigate participant fatigue. 6 participants in the online data repository with incomplete EEG data sets were excluded from forthcoming analyses. This enabled the analyses to be standardized to have exactly the same parameters for each person to support precise comparisons.

To establish how speech-to-language correlates varied with listener attention, we repeated the battery of analyses presented in **Fig 2** whilst modelling either the attended or unattended speech stream. Besides predicting the two speech streams, the only other difference to our first analyses was that we discontinued using the Log-Mel spectrogram because it had previously afforded no predictive benefit (**Fig 2**).

For attended speech, as is illustrated in **Fig 3 Upper Left**, the entire pattern of scalp-average EEG prediction accuracies for the different pairwise model combinations broadly corroborated the earlier finding that the deeper more linguistic Whisper layers were more accurate EEG predictors (**Fig 2**). Specifically, signed ranks comparisons of scalp-average prediction accuracies revealed that combining Whisper L4-6 with Env&Dv elevated prediction accuracies above Env&Dv, and all Whisper layers improved significantly over Lexical Surprisal (See **Fig 3** for test statistics). For example, [Whisper L6 Env&Dv] had a Mean±SEM accuracy of r = 0.055 ±0.003 comparative to Env&Dv (r = 0.041±0.003) where Env&Dv was the most accurate Control. Also echoing **Fig 2**, prediction accuracies increased with layer depth–and speech-to-language transformation. The Mean±SEM Spearman correlation between prediction accuracy and layer depth was 0.78±0.06, which was significantly greater than zero (Signed-rank Z = 4.4, p = 1.2e-5, n = 27, 2-tail).

Follow up electrode-wise analyses that partitioned how Whisper L6, Env&Dv and Lexical Surprisal each contributed to predicting attended speech revealed scalp maps (**Fig 3 Upper Right)** that again echoed **Fig 2**. Whisper L6 dominated responses in bilateral temporal electrodes, Env&Dv also contributed to predicting electrodes in those same scalp locations, and Lexical Surprisal made independent contributions to predicting centroparietal electrodes. The most salient difference to **Fig 2** was that the early Whisper layers (pre-L4) failed to improve over Env&Dv predictions. This could reflect the more challenging listening conditions, or contamination of the EEG signal with traces of acoustic processing of unattended speech.

Differently, and in line with the hypotheses that EEG correlates of speech-to-language transformation would dwindle for unattended speech, no Whisper model of unattended speech improved on the scalp-average prediction accuracies made by Env&Dv (**Fig 3 Lower Left**). Whisper did however complement Lexical Surprisal and this effect was observed for all Whisper layers. Based on our electrode-wise analyses, we are confident that this effect stems from traces of speech acoustics that are residual in Whisper (which is derived directly from spectral speech features) but absent from Lexical Surprisal time-series (which is divorced from speech acoustics aside from tracking word onset times). Specifically, follow up electrode-wise analyses comparing [Whisper L6 Env&Dv Lexical Surprisal] to two-model subsets (e.g. [Whisper L6 Lexical Surprisal]), revealed that Env&Dv was the only representation to uniquely contribute to EEG prediction, and this effect was observed in frontocentral electrodes (**Fig 3**

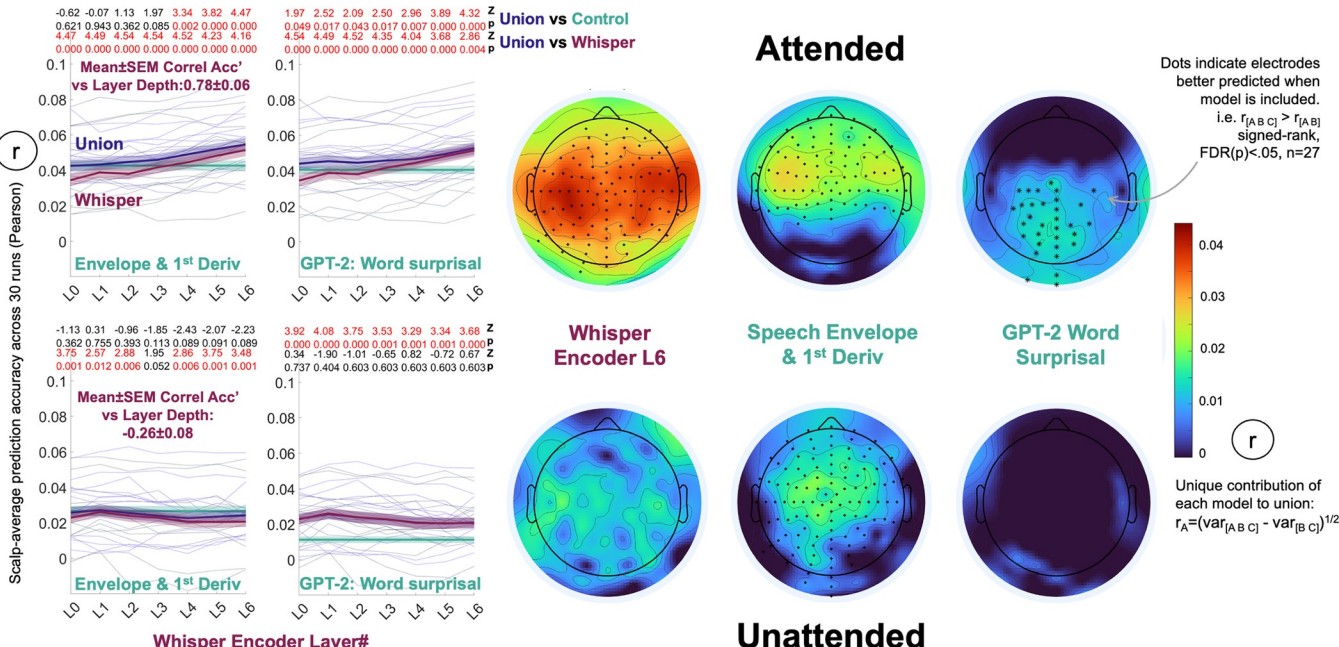

**Fig 3. EEG reflected the linguistic transformation of speech as a function of selective-attention in a two-speaker cocktail party scenario.** Companion results split by story speaker are in **Figs H and I in S1 Text**. **Top Row Attended:** EEG prediction accuracies derived from attended speech models resembled the single audiobook data (**Fig 2**). **Top Left Plots:** Whisper complemented speech envelope-based measures and lexical surprisal in scalp-average EEG prediction. Z and FDR corrected p-values at the top of each plot correspond to signed ranks tests comparing Union EEG prediction accuracies to constituent models. Mirroring **Fig 2**, Whisper's prediction accuracy increased with layer depth and latter layers complemented both the envelope measures and lexical surprisal. **Top Right Scalp Maps:** Electrode-wise analyses and variance partitioning (see **Fig 2** caption for details) revealed that Whisper L6 dominated prediction in bilateral temporal scalp electrodes. Lexical surprisal was reflected in centroparietal electrodes. **Bottom Row Unattended:** EEG prediction accuracies derived from unattended speech models revealed no linguistic contribution. **Bottom Left Plots:** Based on scalp-average measures, Whisper added no predictive value to the envelope-based measures, and rather than increasing, prediction accuracy was weaker in later layers (see signed ranks Z and FDR corrected p-values at the top of each plot). Whisper did however improve on lexical surprisal prediction accuracies (all layers), which is presumably because it still encodes a residual of unattended speech acoustics. **Bottom Right Scalp Maps:** Consistent with the brain processing only superficial acoustic features of unattended speech electrode-wise analyses and predicted variance partitioning echoed that envelope-based measures alone drove prediction over central scalp electrodes.

**Bottom Right**). Thus Env&Dv in isolation provided the most parsimonious EEG model of unattended speech.

Our second selective-attention hypothesis was that when unattended speech was modelled, Whisper layer depth would have little influence on prediction accuracy (the slope across layers in **Fig 2** would flatten). As it turned out, the Mean±SEM Spearman correlation between Whisper prediction accuracy and layer depth across participants was negative: -0.26±0.08, and significantly beneath zero when tested across participants (Signed-rank Z = -2.6, p = 0.01, n = 27, 2-tail). Thus, rather than increasing with layer depth and the transformation from speech-to-language (as for attended speech) EEG prediction accuracy decreased. To provide extra support for this claim, **Table A in S1 Text** presents a linear mixed model analysis of the entire Selective-Attention data set of scalp-average prediction accuracies displayed in **Fig 3**.

In sum, the analyses of cocktail party selective-attention EEG data provide evidence that when listeners disengage from speech–and traditional correlates of lexical surprisal diminish, the predictive advantage afforded by deep rather than early Whisper layers also diminishes in bilateral temporal electrodes. This is consistent with the claim that Whisper recovers electrophysiological correlates of speech-to-language transformation (when speech is attended to at least) rather than providing a brain-like filter of concurrent spectral features (which are still present when speech is not attended).

### Interpreting EEG correlates of speech-to-language transformation

Having consolidated evidence that EEG indexes linguistic transformations of speech, we finally sought to probe the nature of the new EEG signal predicted and in particular estimate the extent to which it reflected lexical processing and contextualization. We focused this investigation on the single audiobook EEG dataset (**Fig 2**), because EEG recordings were longer and not contaminated with dual speech streams unlike the Cocktail Party data set.

### Interpretation: EEG correlates of Whisper partially reflect predictive lexical processing

Because Whisper's last and putatively most linguistic layer was universally the most accurate EEG predictor, we ran further tests to explore how strongly the EEG correlates reflected lexical processing. To this end we first referenced Whisper's EEG predictions to the internal states of a pure language model (GPT-2). To recap, GPT-2 also underpinned the earlier lexical surprisal analysis, and was chosen due to its excellent universal performance in predicting brain data [12,22,24,26,28]. The current analysis differs in its basis on GPT-2's internal states–or layer activations–which are generally thought to encode a mixture of lexical semantics and syntax that help GPT-2 to predict upcoming words. To contrast Lexical Surprisal (**Figs 2 and 3**) reflects the inaccuracy of GPT-2's next-word-predictions (which is the error signal used to optimize GPT-2).

To establish commonalities and complementarities of GPT-2 with different Whisper layers, we predicted EEG with pairwise combinations of GPT-2 L16 and each Whisper layer and then examined how scalp-average prediction accuracy compared to the two isolated constituent models. We selected L16 based on the outcomes of independent research studies computed on fMRI [26], but nonetheless ran post hoc tests with different layers that corroborated the validity of this choice (**Fig J in S1 Text**).

Results are illustrated in **Fig 4A**. Signed ranks comparisons of prediction accuracies revealed that combining GPT-2 with early (L0-4) but not late (L5-L6) Whisper layers boosted accuracy above Whisper alone. Notably, there also were negligible differences in prediction accuracy between Unions of GPT-2 L16 with early and late Whisper layers. For instance the Mean±SEM Spearman Correlation coefficient between [GPT-2 L0-6] scalp-average prediction accuracy and layer depth (L0-6) was 0.14±0.09, which was not significantly greater than zero (Signed rank Z = 0.85, p = 0.4). This suggests that Whisper L5 and L6 share commonalities in lexical representation with GPT-2 that are lacking from Whisper L0 to L4. Critically, this provides evidence that EEG correlates of Whisper L6 indeed partially reflect lexical transformations of speech and also support the notion that Whisper encodes a graded speech-to-language transformation (which we had assumed prior to this analysis).

### Interpretation: EEG preferentially reflects the encoding of 5-10s speech contexts

Because long multi-word speech contexts would seem to be necessary for Whisper to have captured the EEG correlates lexical prediction above, we examined how valuable Whisper's 30s context was for modeling EEG data. To this end we generated Whisper L6 vectors using sliding context windows that we constrained to different durations [.5s 1s 5s 10s 20s 30s] to restrict Whisper's access to linguistic context. As is illustrated in **Fig 4B**, the strongest prediction accuracies were observed for 10s of context, and these accuracies were significantly greater than all other context window sizes shorter than 5s. However, although significant, the gain in prediction accuracy between 0.5s (Mean r = 0.051) and 10s (Mean r = 0.056) was modest, equating to

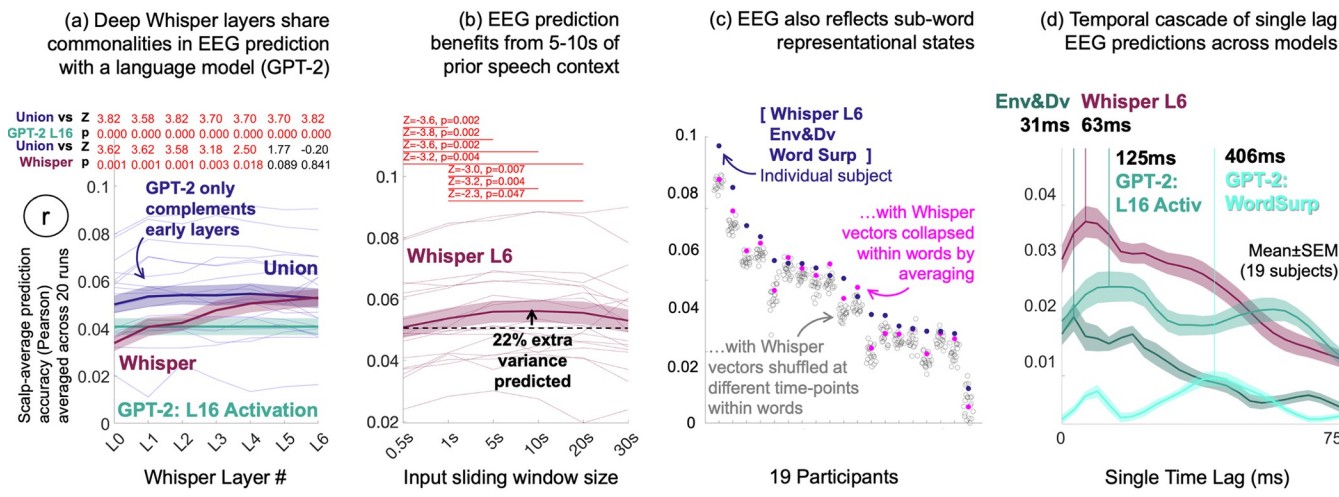

**Fig 4. Interpretation: EEG correlates of speech-to-language transformation reflect a blend of contextualized lexical and sub-lexical representation. (a):** To examine the linguistic nature of Whisper's EEG predictions we referenced them to a pure language model (GPT-2-medium). We focused on GPT-2 L16 based on independent fMRI research ([26], see also **Fig J in S1 Text** for validation of this choice). Consistent with EEG reflecting traces of lexical processing we found that late linguistic Whisper layers captured all variance predicted by GPT-2 (and more), because the Union model (with GPT-2 and Whisper) was no more accurate than Whisper L5 or 6 alone. Differently earlier speech-like layers were complemented by GPT-2. **(b):** To examine whether Whisper's accurate EEG predictions were driven by contextualized representation, Whisper's context window size was constrained to different durations [0.5s, 1s, 5s, 10s, 20s, 30s]. Accuracy was greatest at 5-10s, suggesting that intermediate contexts spanning multiple words were beneficial. Corresponding signed ranks test Z and FDR corrected p-values are displayed on the plot. The dashed horizontal line reflects mean prediction accuracy with a 0.5s context. **(c):** To examine whether Whisper L6's accurate EEG predictions were part driven by sub-lexical structure residual in Whisper's last layer, we disrupted within word structure by either feature-wise averaging L6 vectors within word time boundaries or randomly reordering Whisper vectors within words. The outcome suggested that the EEG data additionally reflected a sub-lexical transformational stage, because either shuffling vectors within words or averaging them compromised EEG prediction in most participants. **(d):** To explore how the relative timing of EEG responses predicted by Whisper compared to the speech envelope and language model, we ran a battery of "single time lag" regression analyses. Model features were offset by a single lag within the range [0 to 750ms in 1/32s steps] and model-to-EEG mappings were fit on each lag separately (rather than all lags), as was repeated for each model in isolation. Whisper preferentially predicted a lag of 63ms, after the speech envelope (31ms) and before both the language model (125ms) and word surprisal (406ms). Note that the illustrated profiles chart (single-lag) prediction accuracies, and as such should not be confused with time-lagged regression beta-coefficients commonly used in the literature to estimate brain temporal response functions.

22% extra variance predicted ($r^2$). A subsequent exploratory analysis of individual electrodes' sensitivity to different context durations (**Fig K in S1 Text**) revealed that all electrodes were preferentially predicted with 5s or more context, and some posterior scalp electrodes were sensitive to lengthier Whisper contexts of 30s.

Critically, these results provide evidence that EEG signals are sensitive to multi-word speech contexts that cannot be modeled by traditional context-invariant categorical approaches. However future work will be necessary to pin down precisely what contextual information was critical. Although speech models appear to encode elements of semantics and syntax [17] that could support lexical and phonological predictions, it is also possible that the contextual advantage draws from cross-speaker normalization, F0 normalization across utterances within speaker, accent stabilization, or consistency of acoustic cues to word boundaries.

## Interpretation: EEG and deep Whisper layers also partially reflect sub-lexical structure

Because the bilateral scalp electrodes best captured by Whisper (**Figs 2 and 3**) are typically considered to reflect low-level speech acoustics and/or categorical speech units [3], and Whisper's current EEG predictions could be partly driven by sub-lexical structure residual in Whisper (as was already hinted by the strong predictions obtained when pairing shallow speech-like whisper layers with GPT-2 in **Fig 4A**), we tested this further. To this end, we evaluated whether

Whisper L6's predictive advantage was entirely driven by lexical representation. If this was the case, we reasoned that either "lexicalizing" L6 by pointwise averaging vectors within word time boundaries, or randomly shuffling the temporal order of vectors within words should have negligible impact on EEG prediction accuracy.

To establish the effects of disrupting within word structure, we lexicalized or shuffled Whisper L6 vectors within words, as described above. We then ran comparative cross-validation analyses, first predicting EEG data with the Union of [Whisper L6 Env&Dv Lexical Surprisal] and then repeating analysis, but replacing Whisper L6 with either its shuffled or lexicalized counterpart. To provide confidence that L6 words with shuffled temporal structure still had some value for predicting EEG, we repeated analyses shuffling Whisper vectors across the entire timeline, which ablated Whisper's unique contribution to prediction altogether.

Consistent with the EEG data also reflecting sub-lexical structure, both experimental manipulations of Whisper damaged prediction accuracy for most participants ([Fig 4C]). Specifically, signed ranks comparisons of scalp-average prediction accuracies between Whisper L6 and lexicalized Whisper L6 revealed a significant drop in the latter (Mean±SEM = 0.059±0.004 and 0.056±0.004 respectively, Z = 2.86, p = 0.0043, n = 19, 2-tailed). When Whisper L6 vectors were randomly shuffled within words, and this process was repeated 20 times, the unshuffled prediction accuracies (0.0534±0.0034, when averaging shuffles with participant) were found to be greatest in 13/19 participants. The cumulative binomial probability of achieving this outcome (p = 1/20) in 13 or more participants is 2.5e-13. We presume that EEG recordings in the other 6 participants reflected correlates of lexical representation coded in Whisper. This could be due to differences in cortical folding between human participants. More specifically, EEG recordings from different people likely contain different relative contributions from different functionally specialized cortical regions, meaning that the EEG of different people could reflect different speech and language representations to varying degrees.

Ablating Whisper's contribution to prediction altogether by shuffling Whisper vectors across the entire timeline (without changing Env&Dv and Lexical Surprisal) produced Mean ±SEM scalp-average prediction accuracies of 0.0380±0.0034 (when repeated 10 times with different random shuffles and averaging within each participant). This was significantly less accurate than predictions derived from within word shuffles (z = -3.8230, p = 1.32e-04, n = 19, signed-rank) suggesting that Whisper still made a predictive contribution after within-word shuffling.

In sum, these analyses suggest that the current EEG correlates of speech-to-language transformation reflect a mixture of both lexical and sub-lexical structure in most participants.

## Interpretation: Whisper best predicts EEG responses that are intermediary between speech acoustics and language

Given the current evidence that EEG responses captured by Whisper reflect lexical and sub-lexical structure we further examined how their timing related to acoustic speech processing and language with the expectation that Whisper would be either intermediary (reflecting a sub-lexical/lexical feature mixture) or would separably reflect both. To explore this, we ran a set of analyses where only a single time-lag of model features was used to predict EEG, rather than all time-lags at once as in our other analyses. Single lags were within the range [0 to 750ms] in 1/32s steps. We reasoned that prediction accuracies derived from different lags would provide an estimate of the EEG response time-delay associated with each model. We were especially interested to see if such an analysis would show a single peak at intermediary lags (indicating an intermediary sub-lexical/lexical feature mixture) or whether it would show a double peak (suggesting separable indexing of sub-lexical and lexical features). Model-to-

EEG mappings were fit on isolated models without variance partitioning to simplify analyses, and because stimulus features that are shared across models might be encoded at different stimulation latencies. In turn, this may exaggerate estimates of models' unique predictive contribution (NB the multi-lag regression analyses presented in the other results do account for this issue). Therefore, for completeness we include a single-lag variance partitioning analysis in **Fig L in S1 Text**.

**Fig 4D** illustrates the EEG response timings preferentially predicted by the different models (scalp-average prediction accuracies). Whisper preferentially predicted EEG at a time delay of 63ms which was intermediary between the speech envelope (31ms) and the language model (GPT-2 L16 activation, 125ms) and word surprisal (406ms), however Whisper's prediction accuracies were also comparatively high across the 400ms time span. This suggests different L6 features may have value for predicting different stages in conversion from sounds to words, and in particular early prelexical representation. A secondary observation was that GPT-2 L16's prediction accuracy profile was doubled humped across response lags, with the second (weaker) peak at 563ms. We speculate the GPT-2 double hump reflects EEG responses associated with consecutive words e.g. the language model at word n both predicts the EEG response to word n and also n+1 (albeit with reduced accuracy).

## Interpretation: How Whisper differs from self-supervised speech models that infer linguistic representations

To gain further insight into the nature of EEG predictions based on Whisper's explicit transformation of speech-to-text, we ran comparative analyses against two self-supervised speech models, that are trained entirely on sound data, with no access to language. For this, we selected Wav2Vec2 [15] and HuBERT [38] that in different studies have provided high accuracy fMRI models [30,31] and a strong basis for decoding model features from MEG and/or EEG data [48,49].

Specifically, like Whisper, Wav2Vec2 and HuBERT deploy Transformer encoders to build contextualized representations of speech, but different to Whisper they are pre-trained to infer the identity of artificially masked speech sounds. As such, the way these models represent speech and language should differ across their layers compared to Whisper. Specifically, past modelling and fMRI research has suggested that the inner layers of self-supervised speech models induce lexical semantic representations from speech [17,31,39]–which could help them to infer contents of masked speech. However, the later layers focus on decoding back to an acoustic speech representation. As such, one might expect a different profile of EEG predictions across the layers of these models compared to the profile we have observed with Whisper–with any such differences adding to our understanding of the Whisper-based EEG predictions. To enable the current results to be cross-referenced to [30,31], we undertook EEG analyses with Wav2Vec2-base and HuBERT-base, which both have 12 layers, and compared them to Whisper-small (which has 12 rather than 6 layers). All networks were run with 30s sliding context widows, and all layer representations were projected onto ten corresponding PCA axes derived from an external dataset (7M, 7F speakers as before).

We found that all three speech models yielded highly accurate scalp-average EEG predictions (**Fig 5**). Whisper L12 was superficially the most accurate (Mean±SEM r = 0.057±0.004), but when compared to the closest runner up Wav2Vec2 L7 (Mean±SEM r = 0.055±0.004), there was no significant difference (Signed-rank Z = 0.55, p = 0.55, n = 19, 2-tail). From visual inspection, the most salient difference between self-supervised models and Whisper was the pattern of EEG prediction accuracies across layers. As expected, whereas Whisper was characterized by an approximately monotonic increase in prediction accuracy tracking with layer

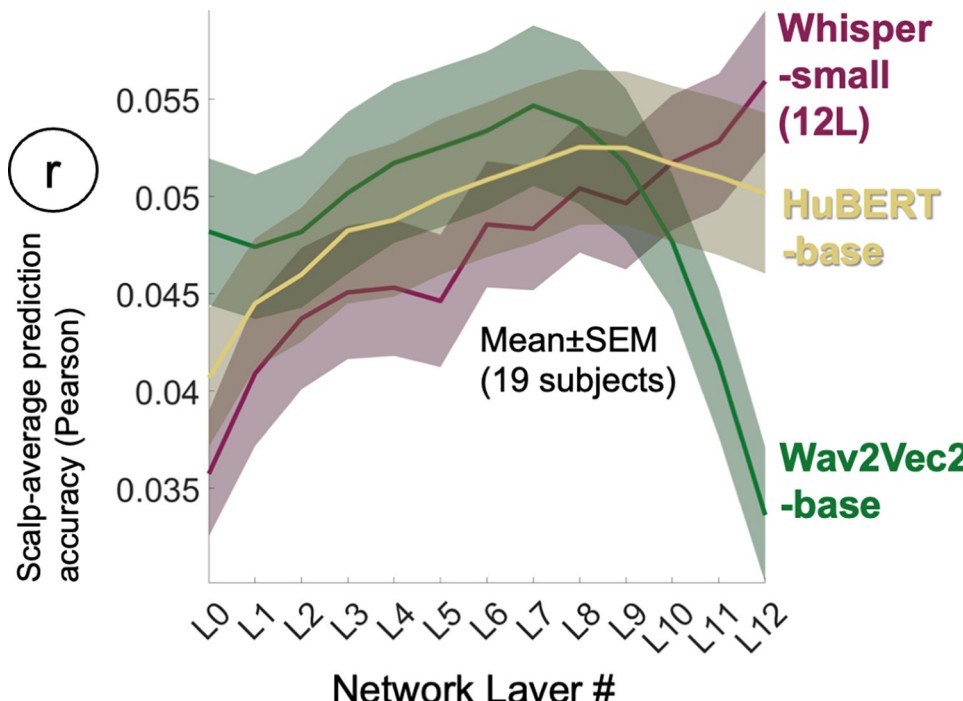

**Fig 5. To explore how Whisper's accurate EEG predictions compared to self-supervised speech models trained without direct access to language (on identifying masked speech sounds) we also repeated analyses with Wav2Vec2 and HuBERT.** To enable cross-referencing to comparative fMRI studies [30,31], we performed this comparison on models comprising 12 layers. Wav2Vec2 and HuBERT both yielded highly accurate predictions but unlike Whisper, the inner layers (L7 and L9 respectively) rather than the last layer were most accurate. Further tests reported in the main text, suggest that Whisper predicted some different components of EEG signal to Wav2Vec2.

depth, it was the intermediate layers of Wav2Vec2 and HuBERT (L7, L9 respectively) that were most accurate. Wav2Vec2 and HuBERT's inner layer predictive advantage observed here on EEG closely mirrors the inner-layer advantages observed in fMRI research [30,31]. Because comparative modelling analyses [39] have specifically linked the representational content of Wav2Vec2 L7-8 to a lexical semantic model (GloVe [50]), we presume this information supports EEG prediction. We further presume that this lexical content diminishes in latter model layers where it is back transformed to make predictions about masked sounds, and this accounts for the associated drop in EEG prediction accuracy.

Finally, to explore for representational commonalities between Whisper L12 and Wav2Vec L7, we predicted EEG with the Union of the two models, and then contrasted accuracies with constituent models. Consistent with Whisper and Wav2Vec2 both contributing to prediction, the Union yielded modestly stronger prediction accuracies (Mean±SEM r = 0.06±0.004, n = 19) than both Wav2Vec2 L7 (Signed-Rank Z = 3.67, p = 2.5e-4, n = 19, 2-tail) and Whisper L12 (Signed-rank Z = 2.37, p = 0.02, n = 19, 2-tail). A variance partitioning analysis suggested

that ~75% of predicted variance was shared between models: $r^2_{Shared} = 100*(r^2_{Whisper}+r^2_{Wav2Vec2}-r^2_{Union})/r^2_{Union})$, ~15% is added by Whisper L12: $[100*(r^2_{Whisper}-r^2_{Shared})/r^2_{Union}]$ and ~9% is contributed by Wav2Vec2 L7: $100*(r^2_{Wav2Vec2}-r^2_{Shared})/r^2_{Union}$.

In sum, these findings reveal that EEG data is also accurately predicted by the more linguistic inner-layers of self-supervised speech models, and inner layers appear to share representational content with Whisper's final most linguistic layer. Whisper's unique contribution to EEG prediction may stem from its direct mapping to language in training. Collectively results are consistent with EEG reflecting a contextualized transformation of speech-to-language.

## Discussion

The current study has revealed electrophysiological correlates of the linguistic transformation of heard speech using the end-to-end speech recognition model Whisper. This addresses a gap in previous work that has typically relied upon hand-crafted context invariant categorical speech units such as phonemes to capture an intermediary phase between sound and words, without modeling the mechanism that maps sounds to categories, and potentially also the representations invoked in this mapping. The current results suggest that: (1) EEG is better predicted with the more complete model provided by Whisper. (2) Whisper helps reveal neural correlates of a contextualized speech transformation that reflects both prelexical and lexical representation and predictive processing which cannot be modeled by context invariant categorical approaches (by definition). To strengthen the case that the newly predicted EEG signal reflected a linguistic transformation as opposed to a brain-like filter of concurrent acoustic speech, the study further demonstrated that Whisper correlates were sensitive to listener attention. Specifically, correlates of Whisper's deeper more linguistic layers selectively diminished comparative to early layers when listeners ignored one speaker in favor of listening to another (for whom the correlates of deep layers were present). More generally, this study exemplifies how deep-learning models can help tackle unresolved questions in human speech comprehension, and in so doing predict neurophysiological data with minimal experimenter intervention.

The flipside of the EEG modelling benefits observed above is that interpreting deep-learning models and what they predict in brain data are both notoriously challenging, because internal representations require empirical investigation to interpret and may neatly not align with spectro-temporal, phonological, lexical and combinatorial components of theoretical models (e.g. [51]). Nonetheless, the inputs, outputs, architecture and training objective(s) of a deep learning model provide initial clues to how information is coded and reshaped throughout the network, and therefore what may drive EEG prediction. To recap Whisper is trained to translate multilingual speech spectra into words in either the input language or English and to also annotate word timing (see also **Methods**). The network architecture deploys a Transformer Encoder to contextualize and transform speech spectrograms into time-aligned vector representations (these vectors were the basis of the current study) and then a Transformer Decoder to transcribe the Encoder output vectors (which we did not examine here, please see **Methods** and **Introduction**). Thus, our starting assumption was that Whisper's Encoder gradually transforms speech into a contextualized linguistic representation–which may be language invariant (perhaps even encoding semantics) because the Encoder output is optimized to prepare speech for transcription either within or between languages. However, at this stage the precise nature of network representations still remained ambiguous.

Having found that the final "most linguistic" layer of Whisper's Encoder provided the most accurate EEG predictions across both electrodes and experiments, it was essential to run further interpretative tests to isolate what drove prediction (see also [29,31]). To probe for

correlates of word processing we referenced Whisper's EEG predictions to a pure language model (GPT-2, **Fig 4A**)—itself a Transformer Decoder—that predicts next-word identity. Consistent with the EEG data encoding features of predictive lexical processing there were commonalities between Whisper and the language model, especially in Whisper's final layer which captured all the EEG variance predicted by the language model, and more. Interestingly, when the language model was paired with the earlier more speech-like Whisper layers, it did add complementary value, and generated EEG predictions on a par with latter Whisper layers. This suggested that EEG both reflects elements of word prediction (as found in GPT-2) alongside pre-lexical speech codes.

In light of the above, we examined the impact of limiting Whisper's opportunity to form predictions, by limiting speech context. Despite our anticipation that EEG prediction accuracy might monotonically improve or asymptote with longer contexts (up to the 30s max), we found 5-10s context to be most accurate (~20% extra variance predicted than .5s). Critically the 5-10s contextual advantage provides evidence that EEG encodes information across multiword speech contexts, and that contextualized models predict components of EEG that context invariant categorical models cannot capture. However, it remains unclear why 5-10s speech is advantageous and future work will be needed to characterize EEG correlates of speech contextualization at lexical and prelexical levels. One approach could be examining how representations in speech models vary in the presence/absence of linguistic contexts that do/don't make upcoming words and sounds predictable (e.g. [14]) and then testing whether model differences capture corresponding differences in electrophysiological responses to the predictable/unpredictable stimuli.

Another potentially revealing approach could be to examine how speech model transcription performance varies when models are optimized on different context lengths, and how this relates to EEG modeling accuracy. Although experimentally retraining Whisper was beyond the scope and means of the current study, we note that [20] ran a comparative language modeling analysis evaluated on fMRI data. In training an LSTM language model on next-word-prediction, they observed negligible word prediction benefit to training with contexts of more than 20 words, and that fMRI prediction accuracy also began to asymptote when models neared a 20-word context. On face value, this 20 word context is consistent with 10s speech, given that one would expect at least 20 words to be spoken in 10s (assuming 120–200 words are spoken per minute [1,52]. However, given that people don't altogether forget what they heard 10s ago, and that large language models now routinely exploit thousands of word contexts [53] and can improve fMRI modeling accuracy [24,29], future work will be required to explore the nature of this correspondence.

To test EEG data for correlates of sub-word content we disrupted the within-word temporal structure of Whisper's final layer which we reasoned should have little effect on EEG prediction if it was driven by single code word representations. The findings were consistent with EEG additionally reflecting sub-word structure, as was evidenced by a modest reduction in EEG prediction accuracy in most participants. However, because EEG has low anatomical resolution, the degree to which this word/sub-word composite reflects intermediary part-speech part-language representational states in the brain, as opposed to a blurred sampling of distinct speech and language-selective neural populations was unclear. More anatomically precise fMRI [29,30,31] and ECoG [32–33] studies suggest that the current EEG correlates of prelexical representation may stem from auditory cortices and in particular Superior Temporal Gyrus where intracranial electrodes are especially sensitive to contextualized phone and syllable representations from HuBERT L10 ([32], see also **Fig 5**). fMRI correlates of language models, and/or candidate lexical representations in speech models have typically been observed to radiate out from auditory cortex over lateral and posterior temporal zones to parietal and inferior frontal cortices [26,29,30,31]) and presumably some of these regions contribute to the current predictive overlap between Whisper and GPT-2, and potentially also lexical surprisal.

Given the availability of different speech models, one might wonder whether Whisper is currently the "best" model of brain activity. We consider that the answer to this question must in part be borne out experimentally through correlations between models and brain data. However, besides this, a study's aims may determine the appropriate model for use. For instance, Whisper's access to a preconfigured cross-lingual text vocabulary renders it an unsuitable model of childhood language acquisition when the word vocabulary also must be learnt (see also [30]). We initially chose to focus on Whisper because it provides an accurate end-to-end transformation of audio to text and may therefore discover intermediary pre-lexical and/or lexical representations from audio data that could help predict brain responses.

However, the current analyses provide no clear answer on whether Whisper is the best EEG model right now. When compared to two self-supervised speech models (Wav2Vec2 and HuBERT)–which have no access to text in training but appear to induce lexical semantic representations in inner layers [31,39]), the top EEG prediction accuracies across layers were equivalent, albeit arising from the inner-layers of self-supervised models, rather than the last layer of Whisper's Encoder. In a variance partitioning analysis, Wav2Vec2 and Whisper overlapped in ~75% of EEG variance predicted. The ~15% extra variance contributed by Whisper might reflect Whisper's access to text in training. However, access to text in training appears not to benefit intracranial electrophysiological models of STG [32]. Relatedly, the inner layers of WavLM [54], a self-supervised extension of HuBERT, were found to mode fMRI data more accurately than Whisper's last layer, at least when accuracy was averaged across all brain voxels [29]. Thus, different speech modeling frameworks recover common representations that predict brain responses and provide convergent evidence that EEG reflects early contextualized linguistic transformation of speech, that are complemented by language models.

To close, we believe the current study helps advance understanding of the neurophysiological processes underpinning speech comprehension and selective attention. This adds to a growing body of research that has reaped the benefits of pre-trained deep-learning approaches for interpreting language or speech [12,20–33]. The current study has helped to broaden this horizon by identifying a self-contained EEG model of speech transformation that sensitively reveals listener attention and suggests how the brain could encode prior speech context in comprehension. Because EEG is both low-cost and widely available, and because the current speech-to-language model is automated, we hope that the approach can also contribute to applied research to help index the linguistic depth of speech processing in developmental and disordered populations.

## Methods

### EEG data and recording parameters

All EEG data analyzed in this study are publicly available and were downloaded from Dryad ([40], https://doi.org/10.5061/dryad.070jc). All EEG data were recorded from 128 scalp electrodes, plus two mastoid channels at 512Hz with a BioSemi ActiveTwo system. To emphasize the low frequency EEG signal that is commonly associated with prelexical and lexical representations ([3,9]), EEG was band-pass filtered between 0.5 and 8Hz using a 3rd order Butterworth filter. Data were down-sampled to 32Hz to reduce the computational burden of forthcoming analyses. In all analyses, EEG data were re-referenced to the average of the mastoid channels.

### Single audiobook EEG experiment and participants

Analyses presented in **Figs 2,4** and **5** were undertaken upon EEG data originally recorded in [3] from 19 participants (aged 19–38 years, 13 male) as they listened to an audiobook recording of The Old Man and the Sea [41] narrated by a male speaker. To mitigate participant

fatigue, EEG recording was split into 20 runs, each of approximately 3mins duration, interleaved with brief breaks. The story line was preserved across the 20 trials, and the first trial corresponded to the start of the story.

## Cocktail party selective attention eeg experiment (Dual Audiobook) and participants

Analyses presented in **Fig 3** were undertaken upon EEG data originally recorded in [55] from 33 participants (aged 23–38 years; 27 male) who were played 30mins of two audiobooks presented simultaneously, but paid attention to only one book. The two audiobooks– 20,000 Leagues under the Sea [46] and Journey to the Center of the Earth [47] were narrated by different male speakers, and were presented via headphones, one to the left ear and the other the right. Participants were requested to selectively attend to one audiobook (in one ear) across the entire experiment. Audiobook presentation was split in thirty runs of 1 min each that were separated by brief breaks to mitigate participant fatigue. We analyzed 15 participants who paid attention to "20,000 Leagues. . ." and 12 who paid attention to "Journey. . .". 6 Participants with incomplete EEG datasets were excluded to enable standardization of all analyses within the nested-cross validation procedure.

## Whisper–A deep-learning model of speech-to-language transformation

Our probe for EEG correlates of speech-to-language transformation–Whisper (Web-scale Supervised Pretraining for Speech Recognition or WSPSR/Whisper, [8])–deploys a Transformer Encoder-Decoder deep-learning architecture [18] that transforms spectral speech features into word transcriptions. Whisper is publicly available and we downloaded the version on Hugging Face [56]: https://huggingface.co/docs/transformers/model_doc/whisper) which was pre-trained to transcribe speech in 99 languages, to translate non-English speech to English words, and to annotate the timing of word boundaries.

Whisper's Encoder and Decoder are both multilayer feedforward networks that each stack multiple layers of so-called Transformer blocks (e.g. L1-6 in Whisper-base, **Figs 2–4**, L1-12 in Whisper-small, **Fig 5**), where each block is itself a multilayer network. The Encoder receives a 30s speech spectrogram as input—an 80 channel Log-Mel Spectrogram, computed on 25ms windows, with step 10ms. The spectrogram is projected into a 512 dimensional space, operating on 20ms time-slices via a convolutional network (L0). Because Transformer layers process speech time-frames in parallel, a sinusoidal positional code is added to each time-frame to indicate its relative position in the sequence. The Transformer layers then refine each 512 dimension time-frame across successive layers into a time-aligned output vector sequence (also 512 dimensions). This output sequence is subsequently fed as a 30s whole into each Decoder layer (as a 1500*512 matrix, with each time-slice corresponding to 20ms). The Decoder iteratively predicts the sequence of words corresponding to the Encoder output, one-by one, conditioned on the Encoder output, with each successive word prediction fed back as new Decoder input. Predicted words are represented as long vectors of probability values, with each entry corresponding to a single word.

The current EEG analyses were exclusively based on Whisper's Encoder module, and we excluded the Decoder to constrain the breadth of our analyses and circumvent the additionally need to manage inaccuracies in Whisper's transcription, especially as relates to word timing. In lieu of this we ran comparative analyses with a pure language model (GPT-2, a Transformer Decoder) of the manually aligned audio book transcript to capture elements of lexical prediction. The only study [33] that we are aware of that has deployed Whisper's Decoder to interpret brain data (ECoG) decoupled the Decoder from the Encoder (such that the Decoder

operated analogously to GPT-2 in processing manually transcribed words, without any access to speech). Indeed, [33] observed no benefit to predicting ECoG data with the original coupled Decoder running with access to contextualized speech data. On these grounds it is unclear whether the Whisper's Decoder would have afforded any predictive advantage over GPT-2 in the current EEG analysis, but it may be a worthy topic of investigation in the future.

We primarily focused analyses on Whisper-base rather than larger Whisper models to reduce the computational burden of our analyses. Larger networks may produce more accurate EEG predictions (e.g. we observed predictions derived from Whisper-small to be more accurate Whisper-base, **Fig 2** vs **Fig 5**).

The current analyses were based on three key assumptions: that Whisper's Encoder transforms speech into representations that are (1) linguistic, (2) contextualized and (3) graded across network layers. Thus, successfully referencing EEG to Whisper would provide evidence that the brain signals reflect contextualized linguistic transformation of speech. Below we detail the bases of these assumptions.

Our assumption that Whisper's Encoder generates linguistic representations was grounded on Whisper's training objectives and architecture. Specifically, we assumed that the Encoder produces linguistic vectors, because they are shaped through optimization to be transcribed into words by the Decoder–optionally into a different language. Thus, Encoder vectors potentially suppress within and between speaker variation in intonation, accents, volume, intonation and so on to encode word identity in a language-invariant and possibly semantic format, at least to the extent these acoustic factors don't compromise the Decoder. However, ultimately the precise linguistic nature of the Encoder vectors is an empirical question and one that we picked up on in our latter Interpretative analyses with reference to a pure deep-learning language model (**Fig 4A**).

The assumption that Whisper produces contextualized speech representations was drawn from Whisper's Encoder-Decoder architecture, which was first introduced to generate contextualized language representations to support machine translation [18]. For instance, without having some context to specify what the ambiguous English word "bat" refers to, the correct French translation would be unclear because French disambiguates the flying mammal (*chauvre souris*—bald mouse) from the sports tool (*batte*). In the case of modelling acoustic speech, context could be helpful to disambiguate noisy or mispronounced sounds, either within or across words. e.g. "television" might be inferred from the mispronunciation "televisiom" and in the case of the "the apple grew on the <noise>", the obscured word/sound is likely to be "tree". Interestingly recent studies of Wav2Vec2 (that unlike Whisper was trained without a language modelling objective) have revealed sensitivity to phonological context [13,14] and lexical identity but not semantic context [14].

Whisper's Encoder's Self Attention mechanism explicitly contextualizes each time-frame, by merging it with other time-frames via feature-wise weighted averaging (See **Fig A in S1 Text** for a detailed illustration, and **Figs 1**, and **B and C in S1 Text** for a visualization of actual network weights). To provide some quick intuition, and taking an example from language, to encode "The vampire bat", one would expect self-attention weights between "bat" and "vampire" to be strong (i.e. "bat" attends to "vampire" and the two become merged). Conversely attention weights between "bat" and "The" may be weak because "The" is not very informative for translating "bat". Intuition aside, the value of contextualized speech encoding for EEG prediction is an empirical question which we address by experimentally constraining Whisper's context (**Fig 4B**).

The third assumption—that Whisper produces a graded multistage transformation (of speech-to-language)—was based on Whisper's multilayer feedforward architecture. Each layer (Transformer block) contextualizes each time-frame via the Self-Attention computation

described above. Because in principle at least, L1-6 produce increasingly contextualized representations which contribute to transitioning speech into a linguistic code (suitable for Decoder transcription), we assumed that this transformation is graded. Nonetheless, we did experimentally test this assumption by referencing the EEG predictions derived from Whisper's different layers to acoustic speech representations (**Fig 2**) and a language model (**Fig 4A**).

For the EEG analyses we extracted 512 dimensional vectors corresponding to each (contextualized) time-frame, that were output from each layer. We focused only on encoder layer outputs rather than within-layer states (e.g. Self-Attention maps) to constrain the breadth of our analyses. As such, the current analyses may have overlooked representational information within Whisper that could be relevant for EEG prediction, and might provide the basis for future studies.

Because Whisper is set up to process 30s speech in parallel–which for the first time-frames in the current experimental stories would enable Whisper to see 30s into the future–we constrained the model to preserve biological realism. This was implemented by feeding the input speech audio waveform (resampled at 16000Hz) into Whisper via a ≤30s sliding window approach, which stepped forward over the waveform in 1/8s steps (corresponding to the 8Hz low-pass filtering of EEG data). Thus, at 10s into a run, Whisper was given access to the past 10s of speech audio waveform, and the series of Whisper vectors within the final 1/8s were saved from each layer of the Encoder and accumulated for the EEG analysis. At 40s into the analysis, Whisper processed 10-40s of audio waveform, and the series Whisper vectors within the final 1/8s were saved from each layer of the Encoder and accumulated for the EEG analysis. Whisper time-series were resampled from 50Hz to 32Hz (to match the EEG data) using the Python module resampy.

In addition, to reduce the computational burden of the EEG Regression Analysis, we applied Principal Components Analysis (PCA) to reduce the 512 dimensional Whisper vectors to 10 dimensions. To mitigate any risk of extracting data-set specific Principal Components and thereby maximize the generality of the approach, PCA axes were derived from an external audiobook dataset (i.e. not used as stimuli in any of the current EEG experiments). The external dataset collated 1.5mins speech from each of 7 males and 7 females narrating one story each (14 different stories in total). We processed each 1.5min dataset through Whisper separately, and extracted activations from L0-L6. We then concatenated each layer's activations for the 14 stories to produce seven separate 512 *(14*1.5mins) matrices. PCA was conducted on each layer to provide 7 sets of PCA axes (L0 to L6). To then reduce the dimensionality of the Whisper datasets used in the current EEG analyses, Whisper vectors were projected onto the first ten Principal Component Axes of the corresponding layer by matrix multiplication. The selection of ten components was ultimately arbitrary, but was our first choice. To sanity check this choice, we ran a set of post hoc analyses with different numbers of components and observed diminishing returns when using twenty or forty components (**Fig D in S1 Text**).

## Wav2Vec2 and HuBERT–self-supervised models of unlabeled acoustic speech

To explore how EEG correlates of Whisper relate to Self-Supervised Speech models that unlike Whisper are pre-trained entirely on unlabeled audio speech (without any access to language), we modeled audio stimuli with Wav2Vec2 [15] and HuBERT [38]. Both Wav2Vec2 and HuBERT are publicly available, and we directly applied the base versions downloaded from https://huggingface.co/docs/transformers/model_doc/wav2vec2 or /hubert respectively.

Architecturally, both Wav2Vec2 and HuBERT share commonalities with Whisper in being contextualized speech models based on deep multilayer Transformer Encoder networks.

However, unlike Whisper neither Wav2Vec2 or HuBERT uses a Transformer Decoder to transcribe Encoder outputs and indeed zero manually labeled data is used in pre-training. In the absence of having language to provide a ground truth for optimization, both Wav2Vec2 and HuBERT are pre-trained to make inferences about the identity of artificially masked speech sounds. For intuition, masking sound segments forces the Transformer blocks (and Self-Attention mechanism, see **Fig A in S1 Text**) to develop contextualized representations of the surround to infer the missing content. To facilitate inferring masked sound identity, both Wav2Vec2 and HuBERT are optimized to generate their own dictionaries of discrete speech sounds, which they approach in different ways (please see the original articles for further details of the differences in both quantization and training objectives). A final difference to Whisper is in audio pre-processing. Both Wav2Vec2 or HuBERT directly extract latent speech features from the audio speech waveform via a convolutional neural network, without spectrogram conversion.

Beyond the network differences, all other speech modeling parameters used for Whisper were carried over: Wav2Vec2 and HuBERT were run using the same ≤30s sliding window approach. Contextualized vector representations of each time-frame were extracted from each network layer (12 layers) and reduced to 10 dimensions via projection onto ten corresponding PCA axes, that had been separately computed for each layer on the same external audiobook dataset used in the Whisper analyses (14 stories narrated by 7 males and 7 females).

## GPT-2 –modeling word prediction and surprisal

To provide a pure language reference against which to interpret Whisper's EEG predictions, we deployed GPT-2 (Generative Pretrained Transformer 2, [19]). GPT-2 is a Transformer Decoder Deep-Learning Model trained entirely upon language to predict the identity of the next word, given a history of up to 1024 preceding words. We selected GPT-2 based on its excellent performance in providing a basis for modelling brain data, spanning fMRI, MEG, ECoG and EEG [12,22,24,26,28].

We deployed GPT-2 in two ways. The first was to capture N400 responses in continuous speech, which are centroparietal negativities following unexpected words that are most pronounced at ~400ms post word onset. Recent research has revealed that estimates of word surprisal (aka lexical surprisal) generated by GPT-2 provide an accurate way to recover the N400 response from continuous natural speech [12]. The second application of GPT-2 was more exploratory in the context of EEG, though has been tested out in studies of fMRI, MEG and ECoG [22,24,26,28]. Here we extracted word activation vectors output from each of GPT-2's layers, and used these as a basis for predicting EEG activity in much the same way as we did for Whisper (please see the previous section).

Both word surprisal estimates and layer-wise word vectors were derived by processing the EEG stimulus story transcripts through GPT-2-medium (a 24 layer Transformer) using a sentence-based sliding window approach: GPT-2 was given as many previous sentences as could fit into its 1024 word context window, and the sliding window was advanced by stepping forward one sentence at a time. Beyond the transcript of the first EEG run, the sliding window straddled across run boundaries (so at the start of run 2, GPT-2 had access to sentences spanning backwards into run 1). We adopted this approach, because participants brains would likewise have had access to this prior story context. With each advance of the sliding window, word surprisal values and layer activations were extracted for all words within the leading (newest) sentence in the sliding window (detailed below).

Word surprisal estimates were computed for each word in the EEG stimulus, except for the very first word in the story (for which there was no preceding word from which to estimate

surprisal). At $word_n$, the estimate of next-word-identity is represented as a long vector of probability values linked to each word in GPT-2's dictionary (this is GPT-2's grand output). Word surprisal was computed as the negative log probability of the actual $word_{n+1}$. To enable the EEG data to be referenced to the series of surprisal values, the surprisal values were aligned to word onset times as "spikes". All other time-points were assigned the value zero.

To reference EEG to GPT-2 word activation vectors, we first harvested layer output vectors from each of the GPT-2's 24 layers. Word vectors from all layers have 1024 dimensions. To reduce the computational burden of forthcoming Multiple Regression analyses, GPT-2 vectors were reduced from 1024 to ten dimensions through projection onto the first ten principal component axes derived from an independent storybook dataset (comprising the first 2250 words from 10 different stories). Ten dimensions were chosen to match the data reduction applied to Whisper. Exploratory analyses (not reported further) suggested there was no substantive advantage to using more than ten dimensions for predicting EEG data. The reduced 10 dimensional vector sequences were time-aligned to EEG, and vectors were stretched to fit to the duration of corresponding words (e.g. if word 3 started at 10s and ended at 10.3s, the vector for word 3 would be aligned to 10s and stretched to span 0.3s). Silent periods (lacking speech) were assigned the value zero.

As an addendum, to simplify the above explanation, we have implied that GPT-2 processes words. However more accurately, GPT-2 processes tokens which can either be words or subwords (which can be useful to model new "out of dictionary" words). For instance, the word "skiff" is treated as two tokens: "sk" and "iff". In such a case GPT-2 would generate two token vectors for one word (or two surprisal estimates). In our analyses, the two token vectors were combined into a single word vector by pointwise summation–and the two token surprisal estimates were likewise summed to provide a single word surprisal estimate.

## Speech envelope and derivative (Env&Dv)–A model of speech audio tracking

To model the EEG correlates of acoustic speech processing, we first computed the speech envelope, which is a time-varying measure of speech signal intensity, integrating across the acoustic frequency bands humans typically can hear. It is now widely accepted that cortical activity reflects the speech envelope [3,57–63]. To compute the envelope, the speech audio waveform (44100Hz) was first lowpass filtered at 20 kHz (22050Hz cutoff frequency, 1 dB passband attenuation, 60 dB stopband attenuation). Then a gammachirp auditory filter-bank was deployed to emulate cochlea filtering [64] by filtering the 128 bands from 80 Hz to 8 kHz with an equal loudness contour. To create a unidimensional time-series, the 128 bands were averaged together.

Based on findings [65,66] that in addition to the speech envelope, cortical responses reflect so-called acoustic onsets–which correspond to positive slopes in the speech envelope–we computed this measure by differencing adjacent elements in the envelope time-series and reassigning negative values with zero.

For all EEG analyses we concatenated the Speech Envelope with acoustic onsets to produce a 2-dimensional time-series, abbreviated as Env&Dv (Dv because the latter measure reflects an approximation of the derivative).

## Log-Mel spectrogram (Whisper's input)

As a further model of EEG correlates of acoustic speech processing, we deployed the 80 channel Log-Mel Spectrogram, that Whisper computes from the audio waveform preprocessing. The Log-Mel spectrogram is a re-representation of the Short-Time Fourier Transform

Spectrogram emphasizing finer frequency resolution for lower frequencies and extracting signal amplitudes across a log-scaled filter bank (with more filters in low frequencies). Whisper's implementation computes the Log-Mel Spectrogram using librosa: https://librosa.org/doc/main/generated/librosa.filters.mel.html, with 80 filters and an upper limit of 8000Hz.

## Mapping models to predict EEG with multiple regression and nested cross-validation

To reference the continuous time-varying EEG signal back to either Whisper or the acoustic and lexical control models we used regularized multiple regression in a nested cross validation framework (**Fig 1**). Multiple regression was deployed to fit a predictive mapping from the time-aligned model/speech representation to each individual EEG electrode (repeated for all 128 electrodes).

To accommodate neural response delays, we temporally offset model/speech time-lines at each of a series of time-lags that stepped from the current (0ms) up to 750ms into the future (which would capture brain responses occurring up to 750ms post stimulation, which we assume spans the period in which speech is transformed to language). The profile of regression beta-coefficients over this 750ms period provides an estimate of the brains' temporal response function (TRF) to stimulus features (See [67] for illustrations). To maintain consistency across all EEG analyses, the same time-lags were used for all models (whether acoustic/speech/linguistic) and combinations thereof.

Model-to-EEG TRF mappings were fit using Ridge Regression on EEG/model data for 18/20 runs. Both EEG and model/speech data sets (18/20 runs) were normalized by z-scoring as is commonplace for Ridge Regression, such that each feature or electrode had zero mean and unit standard deviation. Regression fitting was repeated for each of a range of different regularization penalties which can be considered to mitigate overfitting by squashing and smoothing potential outlier responses in TRF profiles to different degrees. Penalties were: lambda = [0.1 1 1e1 1e2 1e3 1e4 1e5]. The appropriate regularization penalty was estimated as the lambda value providing the most accurate EEG predictions of the 19th "tuning" run, with accuracy averaged across all electrodes. To provide a final estimate of the TRFs ability to generalize to predict new data, the model-to-EEG TRF mapping corresponding to the selected regularization penalty was evaluated on the 20th run. In tests on either the 19th or 20th run, prediction accuracy was evaluated separately for each electrode by computing Pearson correlation between the predicted time-series and the genuine EEG recording (see the circled r on **Fig 1**). Prior to this model, speech and EEG data for runs 19 and 20 were separately feature/electrode-wise normalized by z-scoring (see above). This procedure was repeated, whilst rotating the training/tuning/test run splits to generate separate prediction accuracy estimates for each of the 20 runs. To maintain consistency across all EEG analyses, the same Ridge Regression set up was deployed for all models (whether acoustic/speech /linguistic) and combinations thereof.

To summarize prediction accuracy across runs at each electrode, we computed the electrode-wise mean of correlation coefficients across the 20 runs (Correlation coefficients were r-to-z transformed prior to averaging, and then afterwards, the mean was z-to-r back-transformed by computing arctanh and tanh respectively). Otherwise, the scalp distribution of predicted-vs-observed correlation coefficients was used to provide a coarse estimate of which brain regions encode information found in differentmodel/speech representations (e.g. Figs 2 and 3). Electrode-wise comparisons of different models' accuracy was undertaken using signed ranks tests. P-values were corrected across electrodes for multiple comparisons with False Discovery Rate (FDR [68]).

**Dryad DOI**

10.5061/dryad.070jc.

## Supporting information

**S1 Text. Supplementary figures and table.**
(PDF)

## Author Contributions

**Conceptualization:** Andrew J. Anderson, Chris Davis, Edmund C. Lalor.

**Formal analysis:** Andrew J. Anderson.

**Funding acquisition:** Andrew J. Anderson, Chris Davis, Edmund C. Lalor.

**Investigation:** Andrew J. Anderson.

**Methodology:** Andrew J. Anderson.

**Project administration:** Andrew J. Anderson.

**Resources:** Andrew J. Anderson, Edmund C. Lalor.

**Software:** Andrew J. Anderson.

**Validation:** Andrew J. Anderson.

**Visualization:** Andrew J. Anderson.

**Writing – original draft:** Andrew J. Anderson.

**Writing – review & editing:** Andrew J. Anderson, Chris Davis, Edmund C. Lalor.

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
