## [Decision Letter · Decision Letter 0]

29 Jan 2024

Dear dr anderson,

Thank you very much for submitting your manuscript "Context and Attention Shape Electrophysiological Correlates of Speech-to-Language Transformation" for consideration at PLOS Computational Biology.

As with all papers reviewed by the journal, your manuscript was reviewed by members of the editorial board and by several independent reviewers. In light of the reviews (below this email), we would like to invite the resubmission of a significantly-revised version that takes into account the reviewers' comments. We obtained a higher than average number of expert reviews, likely to do to the strong pertinence of your work to the field. Reviewers 1 and 5 have voiced several critical concerns that must be addressed. Namely, they both are concerned about the current theoretical of the development the work, the lack of appropriate control analyses for the claims made, and the lack of model comparisons with models that use interpretable annotations. These additional analyses and model comparisons would likely necessary in order to convincingly address these concerns. 

We cannot make any decision about publication until we have seen the revised manuscript and your response to the reviewers' comments. Your revised manuscript is also likely to be sent to reviewers for further evaluation.

Sincerely,

Andrea E. Martin, Ph.D.

Academic Editor

PLOS Computational Biology

Daniele Marinazzo

Section Editor

PLOS Computational Biology

Reviewer's Responses to Questions

**Comments to the Authors:**

Reviewer #1: Please see uploaded attachment.

Reviewer #2: Anderson etal report a range of attempts to model EEG responses to naturalistic speech (both single- and multispeaker conditions) with features extracted from off-the-shelf end-to-end deep neural networks trained to process speech sounds, particularly focusing on "whisper", a model trained to transcribe speech to text. They compare the prediction performance to a range of baseline models and find that whisper, especially its latest layers, makes the strongest contribution to predicting hold-out data, while the envelope with its 1st derivative as well as word surprisal also contribute features unavailable from whisper. They then show that these findings are specific to the responses to an attended speaker when presenting multispeaker stimuli. Lastly, they apply a range of tests to interpret the contribution of whisper, focusing on 1) comparisons to the natural language processing model GPT-2 (the last whisper layer fully accounts for GPT-2's contributions), 2) the length of the context provided to whisper (finding that a length of ~10 seconds is best suited to predict EEG responses), 3) the relevance of temporal structure within words (finding that shuffling whisper representations within words harms prediction performance and that the contribution of whisper is thus partly sub-lexical), 4) contributions of individual lags (finding that early lags seem beneficial for acoustic models, mid-range lags seem beneficial for sublexical features such as whisper and long lags seem beneficial for word surprisal effects) and 5) a comparison against self-supervised speech models, finding that none of them can predict EEG better than whisper.

This impressive set of results is paired with an extensive set of visualisations, supplementary figures and highly detailed descriptions of the methods. The authors mainly use careful and suitable analyses and mostly position their research well within the current, quickly developing landscape of related publications. In sum, the authors make it very hard for me to find aspects that could be done differently.

I will here list the result of such efforts:

Title (and abstract)

Within the context of transformer models, it could be helpful to rethink the use of the word "attention" in the title and the abstract (where "listener-attention" is used). I am not a computer scientist, but I feel that this work seems to be relevant beyond the field of cognitive neuroscience and might very well be cited in computer science circles. If the authors manage to find a way to rearrange the title to make it clear that the attention they are referring to is not that of their model, but that of the listeners (I assume this refers to the results in figure 3), this could help with a quick interpretation of the results from skimming its title in reference sections of citing papers to come.

Abstract

It could be helpful to point out here what the optimisation objective of whisper is (textual annotation, as far as I understand). Further, I am unsure whether the use of "accurate" is justified -- after all, the models only explain very small amounts of variance or correlation, and when the audience reads that these models are performing accurately, they might easily get the wrong impression. Critically, I do not think it is necessary to communicate how impressive these results are.

Introduction

1) The present work has similarities to that of Millet and Vaidya. These papers are cited at a later stage, however, I think they could already be cited in the introduction, together with the work of Li et al., nature neuroscience 2023 (https://www.nature.com/articles/s41593-023-01468-4) and potentially the work of Boos et al., NeuroImage 2021 (https://www.sciencedirect.com/science/article/pii/S1053811921003839) which also attempt to model responses to speech with models starting from acoustic features.

2) "such contextual information is not present in purely categorical speech models" -- what exactly is meant by "categorical" here? It seems slightly ambiguous, as I don't see a theoretical reason why categorical models couldn't also include contextual information.

Results

0) In either the introduction or the results section, a clear description of the optimisation objective of whisper should be provided. What does it ultimately predict? Is the loss a cross-entropy over words? Does it build a vector representation of such words in a similar way as word2vec, or does it only model a discrete probability distribution? This would also make it easier to process the later "... that deeper and *more linguistic* Whisper layers were the strongest EEG predictors.". In the methods section, we find "transforms spectral speech features into word representations", but we do not learn about the nature of these word representations.

1) "both in natural listening conditions and when speech is paid attention to or not" -- to me, both of these conditions seem "natural". Maybe one could be called "passive" rather than natural?

2) Reporting the correlation values, but then referring to squared correlation values which "double" seems confusing to me. Might it make more sense to just stick to the correlation values for this section?

3) "The set of correlation coefficients" -- Which set? This refers to the slope over layers?

4) " ... but not the Spectrogram (Whisper's input) uniquely predicted elements of the EEG signal" -- please refer the reader to the corresponding part of figure 2.

5) Figure 2: I might be wrong, but this colourmap looks to me like jet, which has been shown to be suboptimal. A perceptually linear colourmap might be better suited to communicate these results (e.g. viridis, or "turbo" if the authors want to stick to a rainbow scheme).

6) The section reporting results from multispeaker studies could also cite the work by Fiedler et al., NeuroImage 2019 (https://www.sciencedirect.com/science/article/pii/S1053811918320299).

7) "Model features were offset by a single lag" -- this seemed somewhat ambiguous to me -- I assume the authors mean that only a single lag of the features were used to predict EEG rather than the full set. Maybe that is just my reading, but it could potentially be helpful to others to make this more explicit.

Furthermore, this analysis could in principle be carried out in the same style as the other analyses employing variance decomposition, or alternatively the authors could here train the models by excluding the lag in question and then observing whether the performance decreases relative to the full model. Including an invididual lag doesn't tell us whether this particular lag uniquely carried predictive power, and therefore complicates the interpretation.

Discussion

The authors wonder why longer context windows are disadvantageous. It would be interesting to see here how different context lengths affect the performance of whisper on its native optimisation objective -- are 30 seconds more helpful to transcribe than 20 seconds? Ideally, the authors would have access to whisper models trained on different context lengths to generate a 2D representation in the classic style of Yamins et al., 2014 PNAS (Figure 1) or Schrimpf et al., PNAS 2021 (Figure 3). Since this might however be prohibitively cumbersome, the authors might discuss this issue.

Typos:

- responses reflect abstract speech representation as opposed to ... -- in my view, "representation" lacks an s

- This enabled the analyses to be standardized to be have exactly -- "have" should go

- was that we discontinued the using Log-Mel ... -- "the" should go

- EEG Prediction accuracies derived from attended speech models resembled ... -- "Prediction" shouldn't be capitalised

- MEG, ECoG and EEG (Goldsetin et al., 2022) -- should be "Goldstein"?

Reviewer #3: Overall:

The authors present a study investigating the EEG correlates of speech-to-language transformations facilitated by pre-trained deep learning model Whisper. The novelty of this study is in using EEG (instead of fMRI or ECoG) and then investigating the effects of temporal context and attention on predictions (temporal context, in particular, is something other studies have begun looking into, albeit with different models and input modality data).

The study describes the results and interpretations clearly, and the addition of the cocktail party dataset analysis is particularly great in showing that predictions are not resulting from low level acoustic processing. I have no concerns about the analyses themselves.

My concerns are primarily regarding clarity and contextualization. I have major concerns with the introduction and discussion on these issues. Improving on these fronts will, I believe, not only make this report more easily understandable to an audience that either isn’t as familiar with speech and language or isn’t as familiar with correlating deep learning/LLMs with neural activity, but will also strengthen the impact of the current study.

Major:

1- Intro first paragraph: The authors state that categorical models (presumably they are referencing things like acoustic-phonetic STRFs) “cannot capture associated brain activity.” Of course a STRF is not providing a mechanism, but this statement seems naive/misleading as these models are, by definition, fit to reconstruct brain activity. So by definition, it appears to me that they _do_ capture brain activity, they just do not give a comprehensive mechanism. The authors should either clarify why this statement is still true or edit it accordingly.

2- Intro first paragraph statement #2: I think more context is needed on why modelling information not available (such as phonemes) to participants’ brains is limiting. Do the authors believe that phonemes (or whichever speech category) are not part of any intermediate representation? Or do the authors mean to say that this doesn’t give the full picture? The way this is written, it is unclear to me which, and I think that (at least a good portion of) literature supports that phonemes are a useful representation at some level.

3- Throughout the paper (but first mentioned in the intro third paragraph), later layers are referred to as “more linguistic.” This is too vague and I think would benefit from either referencing literature that explicitly tests this, or at the very least some other qualitative explanation. There is some mention of these types of analyses (done by other groups) later in the discussion, but I think it’s important that this is explained up front as much of the analysis is based on this understanding.

4- Intro third paragraph last sentence: this sounds somewhat circular, that they paid attention to one speaker and thus that speaker is represented? I believe the authors are pointing out that purely acoustic representations could be present for both, but only higher level representations exist for the one being attended to. This would benefit from references that suggest that, so that it’s clearer to the reader that not finding this is novel.

5- Overall, I find the introduction extremely lacking in providing context of the field. There are no references to other efforts using deep learning models, which would motivate their use (references to these only come towards the end of the discussion). The authors may find the following review paper on this subject helpful in addition to the many other studies that they do cite (at the very end of the discussion): Jain 2023 https://doi.org/10.1162/nol_a_00101. Having at least a paragraph of discussion on the advantages (or disadvantages) of these approaches would benefit the reader. I also suggest that the authors use this as an opportunity to be specific about how these types of studies can address the called-out limitations of categorical models. This can then be linked to the discussion about how this particular study was able to address some of these limitations and how further work may continue to address remaining gaps.

6- Related to the previous comment, there is also little reference to other studies investigating the effects of temporal context (e.g. another LM-based brain study Jain, Huth 2018 https://proceedings.neurips.cc/paper_files/paper/2018/hash/f471223d1a1614b58a7dc45c9d01df19-Abstract.html) and attention. In particular, the Jain 2018 study is only mentioned vaguely in the discussion along with other papers looking at using LLMs to investigate brain activity. Given that it seems this paper and the Jain study share similar methods/goals, it should be introduced and discussed (e.g. where do these studies differ? Do the results corroborate each other?). It may be that the authors are trying to stick to studies in EEG, which I understand, but since modelling brain activity with deep learning models is so cross-modal (e.g. fMRI, EEG, ECoG studies all coming out), I think this needs to be introduced.

7- Throughout the paper (such as in the introduction, the last paragraph of “Natural Speech EEG Recordings Strongly Reflect the Linguistic Transformation of Acoustic Speech”, and first paragraph of “Interpretation: EEG and Deep Whisper Layers Also Partially Reflect Sub-Lexical Structure”), the authors refer to the concept that the temporal lobe is thought to only process acoustic/spectral features, but I think this is greatly oversimplified. For example, I don’t think many people would say that the STG only processes spectrograms or even just acoustic-phonetic categories (e.g. phonemes). In fact, research from various modalities suggests that things like semantics (e.g. Huth 2016 https://www.nature.com/articles/nature17637) and temporal integration across sub-word categories may be represented here (e.g., this review covers both “sound” information represented in the STG but also points to how the STG may be temporally integrating these types of representations Yi*, Leonard*, Chang Neuron review https://doi.org/10.1016/j.neuron.2019.04.023). I can understand if the authors are referencing low level auditory areas only or only criticizing linear approaches that would be best suited for low-level features, but as it is currently described, this seems to be oversimplified.

8- In the discussion first paragraph, the authors again point out the disadvantage of linear categorical models that do not model the mechanism. However, I don’t see how the authors have necessarily done this either. Their analyses do suggest that representations in Whisper may be represented in the brain, but they don’t analyze the computations in the model or how the brain computes this. Given that, it seems inappropriate to rely on this point for why this paper is novel. I think it would be better to focus on how this study, by using Whisper and the various analyses, are able to probe certain features that these categorical models do not (like the temporal context), which would be important for elucidating exact mechanisms.

9- Given that the authors note that Whisper’s representations might be largely semantic, it seems odd that in the discussion, there’s very little mention of the wide breadth of semantic work (e.g. work from Alex Huth’s lab in particular). For example, I was left wondering whether the semantic maps found in those papers would overlap at all with the findings of this paper. Of course there is a difference in spatial resolution, but it seems that at least a sentence addressing this would place this work better in the context of other brain studies looking at representations related to the speech-to-language transformation.

10- One key question throughout the paper for me was whether the authors believe Whisper is the best model for this or not. For example, if others want to pursue this type of work, is Whisper the only option? I think the authors do a good job of explaining the difference between GPT-2 and Whisper (difference in inputs and training) and highlighting other models (like Wav2Vec2 and HuBERT), but a note on whether any should be used over another would be helpful. Relatedly, if there was a new LLM/deep learning model to come out, how would one decide whether it was useful to use for looking at speech to language transformations?

Minor:

1- Intro first paragraph: Add clarification on what “oracle” models are. Without this definition, the sentence “This evidence has however been challenged…” comes out of nowhere and it’s difficult to understand what this is referencing (e.g. deleting this sentence entirely seems to have no effect as currently written, so if there is an important point here, the authors need to be more clear about it).

2- In Figure 3, it would help to have the labels for which model was used beneath each of the “Unattended” brain panels.

Reviewer #4: In “Context and Attention Shape Electrophysiological Correlates of Speech-to-Language Transformation” Anderson et al. show that a deep learning system called Whisper can be used to generate representations that are particularly good at predicting EEG responses. Moreover, the predictive power increases (almost linearly) with the representations obtained from deeper and presumable more linguistic layers. In such, this study joins a growing body of work that are showing that large language models can be used as a tool to understand, or more candidly, assign functional representations to neurophysiological signals. The paper has many strengths. At the level of the presentation, I found the text to be easy read; it is concise but to the point and with very effective introductory and concluding paragraphs for each result. The figures are also well designed in the sense of being highly informative. The methodology is also rigorous and, again, well explained. Finally, on a more substantive level, there is a significant effort put in the interpretation of the results. These additional analyses reveal, for example, the time scale of the context effects in speech related EEG responses (~10 s) or by manipulating the representations within words that some of the EEG prediction occurred at the sub-lexical level. I also appreciated the fact that Whisper was compared to the unsupervised deep nets, Wav2Vec2 and HuBERT. The comparison is not only useful to show that Whisper provides similar predictive power but because by contrasting representations in the different deep nets (or their training or architecture) one can again gain additional insights on what exactly is being represented in the neurophysiological signal. This manuscript will therefore serve as a nice example on how to effective leverage the power of deep NN in neurolinguistic research. I have a couple of major comments but since I have not analyzed EEG data myself (only fMRI and ECoG), I might be off track.

Major comments.

1. Spatial localization 1: One of the interesting results, which was not extensively developed, is that the GPT-2 word surprisal is not only able to yield additional predictive power (significant for lower layers) but that this additional power is found in a different spatial locations, in the centro-parietal electrodes. I am a bit puzzled by this result because I would expect a transformer based NN to be particularly good at capturing word surprisal as well. It would be interesting to explore this further, maybe by examining the correlations between word surprisal and the Whisper vectors?

2. Spatial localization 2: Given the results of word surprisal, it seems like it would also be interesting to generate electrode maps for all layers of Whisper. You did something like that for GPT-2 in supplemental Fig 5 but not for Whisper. I am guessing that you have tried that but that it was not particularly informative? Please comment.

3. One analysis that it is missing and that I think could also be quite useful is to perform correlations between the vectors of the different layers of Whisper. Alternatively, one could also do a cumulative union and variance partitioning effect to further investigate what is being represented. Maybe doing this analysis and redoing a spatial analysis as suggested in point 2 would reveal interesting results?

Minor comments.

1. I think that it is better practice to use R^2, the cross-validated coefficient of determination, instead of r to quantify prediction in statistical models. Also, if I understand correctly you are showing r values for a single trial (the 20th ) and then average by permutation. I would make this clear in the figure legends as well and say it is a single trial r. This is to your advantage and is useful when comparing results from different studies.

2. In Fig 2, the mean and sem for the control values appear to all be exactly the same for Env, Log-Mel and GPT-2. There must be an error in the code that generated this figure (and hopefully not a cut and paste in illustrator ;-)

3. Some of the methods of Fig2 are repeated in the main text and could probably be eliminated in one or the other.

4. Very minor suggestion: use A,B,C,D for figure labels instead of “mid-left”, “mid-right”.

Frederic Theunissen.

Reviewer #5: The current article is describing an encoding analysis on EEG data of participants listening to natural speech. Specifically they use hidden states of a deep-learning speech recognition system as predictors of neural activity and compare it to a set of baseline models. I think the study is methodologically well executed, but the conclusions are still a bit unclear to me. I have two main points, one that is more severe and concerns what we learn from this study, and the second one more technical.

Phonemes as intermediary representations:

My major problem reading the current paper was to understand its contributions to understanding the “speech-to-language” transformation. The most prominent point that is mentioned e.g in the abstract is that “Contemporary models typically invoke speech categories (e.g. phonemes) as an intermediary representational stage between sounds and words. However, such models are typically hand-crafted and thus do not speak to the neural computations that putatively underpin categorization as phonemes/words.” or in the discussion: “a key limitation of previous work that has typically invoked hand-crafted categorical speech units such as phonemes as an intermediary phase, and thereby neglected modelling the mechanism that maps sounds to categorical units.”

Since phonemes are specifically mentioned as the ‘old way of doing things’ that this paper wants to improve on, I’m surprised that a phoneme encoding model is not at all considered as a comparison in the analyses. To me the nice thing about linguistic categories like phonemes or distinctive features (manner, place) is that they describe a possible intermediate representation that retains sufficient detail such that you can derive words from them (they distinguish minimal pairs). If the brain uses this representation, the neuroscientist can study smaller sub-computations, like how does the brain transform sound into phonemes instead of ‘how does the brain transform sound into words’. I would like to understand what aspect of speech processing the new approach is capturing that previous approaches (e.g. phoneme encoding models) did not. Or whether they are capturing the same aspects but now do it in a more automatized way using a stimulus-computable model. Unfortunately this is not laid out clearly in the paper. Do the authors want to claim: the brain doesn’t use phonemes, but a different intermediary representational format to map between sound and words, one that we cannot describe but is somewhere hidden in the connection weights of the model? Do they want to say that their model captures the ‘mechanisms’ that map sounds to phonemes, i.e. additional intermediary representations? Both claims would require some explicit comparison of Whisper-based to phoneme-based encoding models. Since the last author’s group has used phoneme/feature models on this EEG dataset before (Di Liberto et al., 2015), it is surprising that this was not done.

If it turns out, say, that the Union of individual feature spaces (Envelope/Derivative + Phoneme labels and Word surprisal) predict EEG equally well predicted as the Whisper based encoding model, the authors could still say their model captures all of these features in a stimulus-computable way. I, personally, would think we learn more from considering individual ‘hand-crafted’ feature spaces. But it would make the scope of the study more clear.

Computation of GPT2-based word surprisal:

As described in the Method section, GPT2 does not process words but so-called tokens. From what I understand this tokenization is done prior to GPT2’s training in a data-driven manner, so does not correspond to linguistically meaningful units. The authors take a pragmatic step to aggregate surprisal values to match full words, which makes sense. However, the authors write that they average token surprisal values into a single word surprisal value. I think the correct thing to do would be to sum token surprisal values to get the word surprisal.

What we want is a word’s (w) surprisal conditioned on the previous context (C), which is given by the negative log of it’s contextual probability: -log p(w | C). Now if w is split into two tokens t_1 and t_2, we can get from GPT2 the contextual probability of the first token p(t_1 | C) and the contextual probability of the second token p(t_2 | C, t_1), which is conditioned on context C and the first token t_1. In order to compute p(w | C) we have to compute the joint probability of the two tokens by multiplying the token probabilities p(t_1 | C) * p(t_2 | C, t_1), using the chain rule of probability. So the surprisal should be either calculated as -log ( p(t_1 | C) * p(t_2 | C, t_1) ) or, equivalently, as -( log p(t_1 | C) + log p(t_2 | C, t_1)). In words, the surprisal of the full word in context should be the sum of the two tokens’ surprisal.

Since I have not used GPT2 specifically, I can’t judge how many words this applies to and how much this choice would effect the results. The averaging choice will systematically underestimate surprisal of the tokenized words: if many words are tokenized this could potentially underestimate the predictive value of GPT2-word surprisal in Figure 2. This should be checked and corrected. Of course, if there is a theoretical reason why the authors average rather than sum surprisal values, they should explain, but I can’t see a good reason.

Minor points:

Description of Whisper. Although the Whisper architecture is described in detail in the Methods, some of the important aspects should be mentioned in the main text. Currently the Introduction and Results section describes it as a model that transforms speech into “language” (There are no line numbers in the text so it’s difficult to refer to specific text, but this is repeated multiple times). This feels a bit imprecise: what is meant by language? Phonemes? Words? Meaning? It makes it hard to understand what the authors mean by the “later, more linguistic layers” without jumping to the Methods first. As I understand, Whisper is trained to generate word-level transcriptions using a separate decoder, even translating to a different language. This is important information, since it means that the later layers should represent very high-level aspects of language.

**Have the authors made all data and (if applicable) computational code underlying the findings in their manuscript fully available?**

Reviewer #1: None

Reviewer #2: **No: **Code will only be released upon publication -- I can thus not comment on the code (yet).

Reviewer #3: **No: **Datasets are public, code only at publication (so not yet).

Reviewer #4: Yes

Reviewer #5: None

PLOS authors have the option to publish the peer review history of their article (what does this mean?). If published, this will include your full peer review and any attached files.

Reviewer #1: No

Reviewer #2: **Yes: **Christoph Daube

Reviewer #3: No

Reviewer #4: **Yes: **Frederic E Theunissen

Reviewer #5: No
---

## [Decision Letter · Decision Letter 1]

27 Aug 2024

Dear dr anderson,

Thank you very much for submitting your manuscript "Deep-Learning Models Reveal Context and Listener Attention Shape Electrophysiological Correlates of Speech-to-Language Transformation" for consideration at PLOS Computational Biology. Your revision clarified most points, with a couple of relevant exceptions raised by the first reviewer which need to be addressed.

You should also change the colormap choice. The perceptual bias is a fact and should not be ignored or dismissed by eyeballing arguments. Furthermore you use the same map both in sequential and diverging settings, and this is very confusing. Here https://nl.mathworks.com/matlabcentral/fileexchange/68546-crameri-perceptually-uniform-scientific-colormaps you can find everything you need.

Sincerely,

Daniele Marinazzo

Section Editor

PLOS Computational Biology

Daniele Marinazzo

Section Editor

PLOS Computational Biology

Reviewer's Responses to Questions

**Comments to the Authors:**

Reviewer #1: Please see uploaded attachment.

Reviewer #2: The authors have responded well to all my points. I have no further comments.

Reviewer #3: I appreciate the authors’ detailed response to my comments. They have sufficiently addressed my concerns and I believe the revised text has substantially improved. I have no further comments and recommend this manuscript for publication.

Reviewer #4: Thank you for addressing all of my comments. The comments from the other reviewers were also thoroughly addressed.

Reviewer #5: I thank the authors for their revisions to the paper. I think this made the contribution a bit clearer to me, especially in light of the previous paper by Daube et al. (2019) that showed that the EEG explanatory power of phoneme categories can be more parsimoniously captured by acoustic edges. This was an important paper, but a bit frustrating: are acoustic edges the only thing that can be captured using EEG? What I find important in the current paper is that it shows (on the same data) that EEG captures indeed more about the sublexical transformation. I see this as an insight about the data that we have available, less as an insight about speech processing in the brain. Now the hard part (to derive a “more complete brain model”, as the authors say) is to figure out what drives the unique variance captured by the deep learning model. Maybe it is again driven by some simpler features that the authors failed to control for here (and maybe we’ll find out in Daube, 2025..?). Maybe it shows that the dynamics of disambiguating linguistic categories in real-time is important and is something that a simple model of phoneme category labels does not capture (as the authors show, there’s not much unique explained variance over and above the Whisper features). What I take away from it is not that linguistic categories like phonemes don’t matter (there’s a lot of evidence from linguistic theory and psycholinguistics) but that it’s much harder to link them to neurophysiological data than expected.

**Have the authors made all data and (if applicable) computational code underlying the findings in their manuscript fully available?**

Reviewer #1: None

Reviewer #2: Yes

Reviewer #3: Yes

Reviewer #4: Yes

Reviewer #5: Yes

PLOS authors have the option to publish the peer review history of their article (what does this mean?). If published, this will include your full peer review and any attached files.

Reviewer #1: No

Reviewer #2: **Yes: **Christoph Daube

Reviewer #3: No

Reviewer #4: **Yes: **Frederic Theunissen

Reviewer #5: No

Figure Files:

Data Requirements:

Reproducibility:

References:

---

## [Editor Report · Decision Letter 2]

4 Oct 2024

Dear dr anderson,

We are pleased to inform you that your manuscript 'Deep-Learning Models Reveal How Context and Listener Attention Shape Electrophysiological Correlates of Speech-to-Language Transformation' has been provisionally accepted for publication in PLOS Computational Biology.

Best regards,

Daniele Marinazzo

Section Editor

PLOS Computational Biology

Daniele Marinazzo

Section Editor

PLOS Computational Biology

---

## [Editor Report · Acceptance letter]

30 Oct 2024

PCOMPBIOL-D-23-01822R2 

Deep-Learning Models Reveal How Context and Listener Attention Shape Electrophysiological Correlates of Speech-to-Language Transformation

Dear Dr anderson,

I am pleased to inform you that your manuscript has been formally accepted for publication in PLOS Computational Biology. Your manuscript is now with our production department and you will be notified of the publication date in due course.

With kind regards,

Anita Estes
